

# Impact of assimilating sea ice concentration, sea ice thickness and snow depth in a coupled ocean-sea ice modeling system

Sindre Fritzner[1], Rune Graversen[1], Kai H. Christensen[2], Philip Rostosky[3], and Keguang Wang[4]

[1]UiT The Arctic University of Norway, Tromsø, Norway
[2]The Norwegian Meteorological Institute, Oslo, Norway
[3]Institute of Environmental Physics, University of Bremen, Germany
[4]The Norwegian Meteorological Institute, Tromsø, Norway

*Correspondence to:* Sindre Fritzner (sindre.m.fritzner@uit.no)

**Abstract.** The accuracy of the initial state is very important for the quality of a forecast, and data assimilation is crucial for obtaining a best possible initial state. For many years, sea-ice concentration was the only parameter used for assimilation into numerical sea-ice models. Sea-ice concentration can easily be observed by satellites, and satellite observations provide a full Arctic coverage. During the last decade, an increasing number of sea-ice related variables have become available, these include sea-ice thickness and snow depth, which are both important parameters in the numerical sea-ice models. In the present study, a coupled ocean-sea-ice model is used to asses the assimilation impact of sea-ice thickness and snow depth on the model. The model system with the assimilation of these parameters is verified by comparison with a system assimilating only ice concentration and a system having no assimilation. The observations assimilated are sea ice concentration from the Ocean and Sea Ice Satellite Application facility, thin sea ice thickness from the European Space Agency's (ESA) Soil Moisture and Ocean Salinity mission, thick sea ice thickness from ESA's CryoSat satellite, and a new snow depth product derived from the National Space Agency's Advanced Microwave Scanning Radiometers (AMSR-E/AMSR-2) satellites. The model results are verified by comparing assimilated observations and independent observations of ice concentration from AMSR-E/AMSR-2, and ice thickness and snow depth from the IceBridge campaign. It is found that the assimilation of ice thickness strongly improves ice concentration, ice thickness and snow depth, while the snow observations have a positive effect on snow thickness and ice concentration. In our study, the seasonal forecast showed that assimilating snow depth lead to a worse estimation of sea-ice extent compared to the other assimilation systems, the other three gave similar results. The improvements due to assimilation were found to last for at least 3-4 months, possibly even longer.

## 1 Introduction

Observations show that for the last 50 years there has been a decline in both Arctic sea-ice extent (Stroeve et al., 2007; Perovich et al., 2017) and sea-ice thickness (Kwok and Rothrock, 2009), in addition, models show that the sea-ice decline is likely to





continue (Zhang and Walsh, 2006). Wang and Overland (2012) estimate the Arctic ocean to be nearly ice-free within the 2030s. This large change in the global climate system leads to a need for improved models and forecasting systems due to more variable and mobile Arctic sea ice (Eicken, 2013). In addition, a decreased amount of sea ice will lead to increased Arctic ship traffic (Smith and Stephenson, 2013). Safe travel in the Arctic is dependent on accurate knowledge of weather and sea ice.

The Arctic is characterized by harsh conditions involving for instance sea ice, icebergs, and polar low storms. The numerical weather prediction models are becoming more complex and detailed, but still, the vital part of an accurate forecast is the model initial state. Accurate initial states can be achieved by assimilating observations into the model system.

For sea-ice modelling in the Arctic, observations are sparse. The sea-ice concentration (SIC), defined as the fraction of the total area covered by sea-ice, has been available since the start of the satellite era in 1979, but observations of other parameters

such as sea ice thickness (SIT) are more difficult to obtain because of the remote location, and satellites cannot easily be used to extract information about the SIT. The passive microwave satellites derive SIC from brightness temperatures, but many of the earth observing satellites do not have sufficient wavelength to observe changes in the brightness temperature as a function of the SIT. Thus acquiring SIT from satellites are significantly more difficult than SIC, but as will be described later satellites using the L-band frequency can, to some degree, be used to measure the SIT as a function of brightness temperature.

During the last 15 years, there have been various studies of SIC assimilation, using several different models and assimilation methods. Lisæter et al. (2003) assimilated SIC obtained from passive microwave satellite into a coupled ocean-ice model using the Ensemble Kalman Filter (EnKF; Evensen, 1994; Burgers et al., 1998). In the study of Lisæter et al. (2003), the assimilation was found to have a strong effect on the modelled SIC, and also small effects on other model parameters due to the multivariate properties of the EnKF. The multivariate properties of the EnKF consist of a model update for all model variables based

on correlation with the observed variables. A similar SIC assimilation study using the 3D-Variational (3D-Var) assimilation method was done by Caya et al. (2010). In this study, both ice charts from the Canadian east coast and Radarsat 2 SIC observations were assimilated. Significant improvements to the short-term forecast were found for the assimilation system. Studies with the coupled ocean-ice model TOPAZ (Sakov et al., 2012) have shown improvements to SIT and multivariate effects on SIT for assimilation of SIC (Sakov et al., 2012). Other SIC studies have been done by Lindsay and Zhang (2006)

and Wang et al. (2013) both using nudging methods to show model improvements for SIC assimilation.

In recent years there has been a focus on increasing the number of observable ice parameters, especially accurate knowledge of the Arctic SIT is important in order to quantify changes in the total Arctic sea-ice volume and to elucidate changes related to for instance global warming. Dedicated satellite altimeters like ICESat (Forsberg and Skourup, 2005) and CryoSat-2 (Laxon et al., 2013) have been prepared for SIT measurements. These satellites use measurements of the ice freeboard to calculate

the SIT (Kurtz and Harbeck, 2017; Kurtz et al., 2014b). Another source of satellite SIT observations is the European Space Agency's (ESA) Soil Moisture and Ocean Salinity (SMOS) mission. The SMOS mission uses L-band passive microwave measurements utilizing long penetration depth and a relationship between observed brightness temperature and ice thickness (Tian-Kunze et al., 2016). However, in general, the uncertainties of the CryoSat and SMOS SIT observations are high (Zygmuntowska et al., 2014; Xie et al., 2016), which result in reduced, though still valuable, observational information available

for assimilation into the model system. The SIT observations are limited to winter conditions when the snow and ice are dry.



One of the first studies with SIT assimilation was done by Lisæter et al. (2007). In this study, computer-generated SIT observations simulating CryoSat observations were assimilated into a coupled ice-ocean model using the EnKF. The assimilation showed significant effects on the model state; both improvements to the modelled SIT and multivariate effects on SIC, ocean temperature and ocean salinity were found. Yang et al. (2014) used the localized singular evolutive interpolated Kalman filter (Pham, 2001) to assimilate the SMOS SIT observations into the Massachusetts Institute of Technology general circulation model (Marshall et al., 1997). In this study, an improved thickness forecast when assimilating SMOS observations and some improvements to the SIC forecasts were found. Similar to Yang et al. (2014), Xie et al. (2016) used the EnKF to assimilate SMOS SIT observations into the TOPAZ system (Sakov et al., 2012). In this study it was found that assimilation of SMOS observations showed improvements for the ice thickness along the ice edge, both compared to SIT observations not assimilated and compared to the SMOS observations themselves. In general, similar to that found by Yang et al. (2014) the SMOS observations were found to have a relatively small impact on the SIC, and the SIT far from the ice edge.

Fritzner et al. (2018) assimilated SMOS observations into a standalone sea-ice model with the EnKF. This study showed that due to the correlation between SIC and SIT, the SMOS observations were found to have a positive effect on the modelled SIC. Chen et al. (2017) assimilated both the SMOS thin SIT and the CryoSat thick SIT into the National Centers for Environmental Prediction's (NCEP) Climate Forecast System version 2 (Saha et al., 2014) using the localized error subspace transform ensemble Kalman filter (Nerger and Hiller, 2013). This study showed improved sea-ice prediction with SIT assimilation. Thus verifying the importance of SIT observations to achieve accurate sea-ice forecasts.

Recent attempts have proved that it might be possible to observe snow depth from satellite (Markus and Cavalieri, 1998; Maaß et al., 2013; Rostosky et al., 2018). Both Maaß et al. (2013) and Rostosky et al. (2018) used a relationship between observed brightness temperature and snow depth to calculate the latter variable. Due to the close connection between snow, albedo and ice melting, accurately modelled snow depths are expected to have a large impact on the snow and ice models. Snow observations are limited to the winter season when the ice and snow are dry.

In our study, a coupled ocean sea-ice model (Kristensen et al., 2017) is used. The coupled model is prepared for improved sea-ice representation compared to previous coupled ocean sea-ice models. This improvement will give a deeper insight into how sea-ice is affecting both the ocean and atmosphere. The assimilation system will be tested with different kinds of observations to analyse both long-term and short-term effects. Observations of SIC, SIT and snow depth are assimilated. The results will be verified with independent and semi-independent data, in addition to forecasts both in summer and winter.

This study is important in order to elucidate the effect of different sea-ice observations and reveal the most important observations for an improved sea-ice forecast. Even though some studies have looked into the assimilation of different SIT observations, as far as we know this is the first study to compare the effect of the different observations on the assimilation system. In addition, as far as we know, this is the first study to present the assimilation of snow depth observations in a coupled ocean sea-ice model.





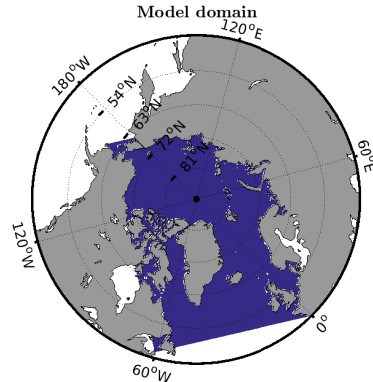

**Figure 1.** The model domain, the blue area is covered by the model, grey area indicate land areas

## 2 The coupled ocean sea-ice model

The coupled model (Kristensen et al., 2017) is based on the Regional Ocean Modelling System (ROMS; Shchepetkin and McWilliams, 2005; Moore et al., 2011) version 3.6 as the ocean component and the Los Alamos sea-ice model version 5.1.2 (CICE; Hunke and Dukowicz, 1997; Hunke et al., 2015a) as the ice component. The ROMS model is a state-of-the-art ocean
model, which in our study is configured with 35 terrain-following vertical layers. The eddy viscosity and eddy diffusivity are parametrized using a second-order turbulence closure model.

    The CICE model is a state-of-the-art sea-ice model with 5 thickness categories, 7 ice layers and one snow layer. The model has a thermodynamic component calculating the local growth rate of snow and ice, ice dynamics component calculating ice drift based on the material ice strength, a transport component and a ridging parametrization used to distribute ice in thickness
categories (Hunke et al., 2015a). The model has a horizontal resolution of 20 km with 242x322 grid cells covering the entire Arctic ocean. The model domain covering the Arctic sea is shown in Fig. 1.

    The coupled model is forced by atmospheric data from the ERA-Interim dataset from the European Centre for Medium Ranged Weather Forecast (ECMWF; Dee et al., 2011). In addition, the model has prescribed ocean boundary and climatic forcing from the Fast Ocean Atmosphere Model (FOAM; Bell et al., 2003). The assimilation system used in the model is
the Ensemble Kalman Filter. The code used for assimilation is the EnKF-c code (Sakov, 2015). The EnKF-c is an easy-to-implement and efficient framework for off-line data assimilation for use in geophysical models.

## 3 Observations

In the present study, observations related to the Arctic sea-ice are used for assimilation, these include SIC, SIT and snow depth. The SIC observations used for assimilation are from the European Organisation for the Exploitation of Meteorological
Satellites (EUMETSAT) Ocean and Sea Ice Satellite Application Facility (OSISAF; Tonboe et al., 2016). The SIC product is the near-real-time global sea-ice concentration product. This dataset contains SIC observations calculated from brightness



temperatures measured by SSMI/S passive microwave. The SSMI/S brightness temperatures are corrected for air temperature, wind roughening over open water and water vapour in the atmosphere by the ECMWF numerical weather prediction (NWP) model (Andersen et al., 2006). To convert from brightness temperatures to SIC a combination of the Bootstrap and the Bristol Algorithm is used (Tonboe et al., 2016). The Bootstrap algorithm is primarily used for observations with low SIC, and the

Bristol algorithm for high SIC. The older OSISAF products do not include an error estimate, but an estimate of the observation confidence. The observation confidences are a simple measure of the observations quality, where 5 is excellent quality, 2 indicate poor quality, 1 indicates computation failure, and 0 no data. In the more recent OSISAF observations, a total uncertainty parameter is associated with each observation. In our study, the observation uncertainty of the OSISAF observations where given by the following formula:

$$TU = a + b * (5 - C), \tag{1}$$

where $C$ is the confidence and $TU$ is the total uncertainty, $a = 0.06$ and $b = 0.1$ are estimated based on the relationship between confidence and uncertainty in the more recent OSISAF observations. Observations flagged with a confidence of 0 or 1 are not used in our study. For verification of the modelled SIC, the AMSR-E/Aqua Daily L3 12.5km Sea Ice Concentration product was used (Cavalieri et al., 2014), consisting of satellite observations from the National Space Agency's Advanced Mi-

crowave Scanning Radiometers (AMSR-E/AMSR-2). The AMSR-E/2 observations are, as the OSISAF SIC observations, also based on measurements from a passive microwave measuring the brightness temperature, and the observations are structured on a 12.5km grid. For the AMSR-E/2 dataset, the Enhanced NASA Team algorithm (Markus and Cavalieri, 2000) is used to convert from brightness temperatures to SIC. The OSISAF and AMSR-E/2 datasets are different data products, but are in many cases tuned to give similar results and cannot be viewed as true independent datasets. The AMSR-E/2 product has a gap from

October 2011, when AMSR-E failed, until AMSR-2 became operational in July 2012, this is in the middle of our analysis period resulting in less data for verification.

Two different SIT products are assimilated. For thick SIT observations, the CryoSat-2 Level-4 Sea Ice Thickness product is used (Kurtz and Harbeck, 2017). The CryoSat observations are based on radar altimeter measurements of sea ice freeboard. The SIT is derived assuming nominal densities for ice, snow and water, and only valid for high concentration ice (> 70 %;

Kurtz et al., 2014b), thus they are assumed to be observations of thick ice relative to the SMOS observations. The snow depth used to calculate sea-ice elevation is constructed from the Warren climatology of snow depth (Warren et al., 1999), modified to account for the loss of multi-year ice in recent years (Kurtz and Farrell, 2011). The dataset has a spatial resolution of 25 km and a 30-day average temporal resolution covering the entire Arctic. For the CryoSat dataset, no uncertainty estimates are provided, thus following Zygmuntowska et al. (2014) an uncertainty of 0.5 m was used for all CryoSat observations. Due to

the low temporal coverage, this is most likely an underestimation of the uncertainty, and other publications have suggested higher uncertainties (Xie et al., 2016; Chen et al., 2017). In our study, the main focus is on the impact of the observations on the assimilation system and thus a low error is applied in order to elucidate the model impact of the observations. Since the Cryosat dataset is only valid for high concentration ice, all observations are in the internal part of the Arctic sea ice, and will



in future reference also be referred to as internal ice thickness. The Cryosat observations are only available in the cold season from October to April.

For thin SIT observations, the daily L3C SMOS Sea Ice Thickness version 3.1 is used (Tian-Kunze et al., 2016). These SIT observations are acquired from a satellite using a passive microwave with L-band frequency. Measurements of brightness

temperatures are converted into SIT using a radiation and thermodynamic model based on penetration depth (Tian-Kunze et al., 2014). Xie et al. (2016) found that observations thinner than around 0.4 m were the most realistic to use in the analysis, hence in this study observations thicker than 0.5 m have not been used. For the SMOS observations it is assumed that all observations are acquired at 100 % SIC, thus the observations are assimilated as normalised ice volume. The SMOS dataset has a resolution of 12.5 km and is structured on a stereographic grid. Since all SMOS observations are thinner than 0.5 m they are all located

in the vicinity of the Arctic ice rim, and will in future reference also be referred to as rim ice thickness. As for the internal ice thickness observations, the SMOS SIT are only available in the cold season from October to April.

For verification of the modelled SIT, the weekly combined SMOS-CryoSat dataset version 1.3 was used (Ricker et al., 2017). This observation product provides SIT observations covering the whole Arctic during the cold season. In addition, the IceBridge L4 Sea Ice Thickness observations are used for verification (Kurtz et al., 2013, 2014a). This dataset consists of

SIT and snow depth measurements from an aeroplane, using a radar altimeter measuring the ice freeboard. The IceBridge observations are limited temporally to March-April, and spatially to parts of the Beaufort Sea, the Canadian Archipelago, and north of Greenland.

The snow depth observations are derived from AMSR-E/2 observed brightness temperatures (Rostosky et al., 2018). The data are available on a daily basis with a resolution of 25 km x 25 km. The algorithm uses the same technique which was

developed by Markus and Cavalieri (1998) to retrieve snow depth over Antarctic sea ice. Their product is based on an empirical relationship between the gradient ratio of the 37 GHz and 19 GHz brightness temperature observations and Antarctic snow depth. It was adapted to retrieve snow depth on Arctic sea ice (Comiso et al., 2003), but due to the radiometric properties of Arctic multi-year ice, the retrieval is limited to first-year ice only. The new product by Rostosky et al. (2018) makes use of lower frequency channels (i.e. brightness temperature observations at 6.9 GHz) which are less sensitive to the Arctic multi-year

ice and thus the retrieval can be, with some exceptions (Rostosky et al., 2018), applied over the whole Arctic sea ice. The new snow depth retrieval was trained and evaluated using NASA's Operation Icebridge airborne snow depth observations (Newman et al., 2014). Those observations are, however, mainly limited to March and April and, so far, no evaluation of the snow depth product exists for the remaining winter season. We, therefore, limit our analysis to snow depth observations in March and April. When the model simulations were performed the snow depth product was in its early development state. By now, a slightly

updated version of the snow depth product exists, but since the overall differences between the updated version and the early state version are small we do not expect the updated data set to yield substantially different results.





## 4 Methods and model setup

### 4.1 The Ensemble kalman filter

The Ensemble Kalman Filter (EnKF) is a sequential data-assimilation method used in a wide variety of geophysical systems (Evensen, 1994, 2009; Houtekamer and Zhang, 2016). The analysis equation for the EnKF is given by (Jazwinski, 1970; Evensen, 2003),

$$\mathbf{x}_a = \mathbf{x}_b + \mathbf{P}_b\mathbf{H}^T \left( \mathbf{HP}_b\mathbf{H}^T + \mathbf{R}\right)^{-1} (\mathbf{y} - \mathbf{Hx}_b). \tag{2}$$

The model background and analysis state vectors are matrices given by, $\mathbf{x}_b \in \mathbb{R}^{n \times N}$ and $\mathbf{x}_a \in \mathbb{R}^{n \times N}$, respectively. Here $n$ is the number of variables (that will become updated) times number of grid cells, and $N$ is the number of ensemble members. The covariance of the observations is given by $\mathbf{R} \in \mathbb{R}^{m \times m}$, where $m$ is the number of observations, $\mathbf{H} \in \mathbb{R}^{m \times n}$ is the observation operator, which is a transformation operator between model and observations space, and $\mathbf{y} \in \mathbb{R}^{m \times N}$ is the observation matrix. For the EnKF the background error covariance matrix, $\mathbf{P}_b$, is estimated based on the covariance of an ensemble of model states. The ensemble is generated by either perturbing the forcing, the model parameters, the observations or a combination of the three. The estimator for model error covariance, $\mathbf{P_b} \in \mathbb{R}^{n \times n}$, is

$$\mathbf{P_b} = \overline{((\mathbf{x}_b - \overline{\mathbf{x}_b})(\mathbf{x}_b - \overline{\mathbf{x}_b})^T)}. \tag{3}$$

The overbars indicate an ensemble average. In our study, the Deterministic Ensemble Kalman Filter (DEnKF) proposed by Sakov and Oke (2008) is used. This method solves the analysis equation without the use of perturbed observations.

When using the EnKF spurious co-variances might occur due to distant state vector elements and insufficient model rank when small ensemble sizes are used. These artefacts can be reduced by using a method for localization (Evensen, 2003; Sakov and Bertino, 2011), limiting the assimilation to affect a smaller area. There are several methods for localization, and in this study, the polynomial taper function (Gaspari and Cohn, 1999) is used. The taper function is a bell-shaped function providing stronger influence to nearby grid cells.

### 4.2 Ensemble spread

Sufficient ensemble spread is essential for a robust and well-functioning EnKF assimilation system. In general, this is maintained by the Kalman Filter equations, but it is important to also take into account the uncertainty in the model and the atmospheric forcing. The atmospheric forcing is perturbed to account for uncertainty in the forcing. The atmospheric forcing is perturbed using smooth pseudo-random fields (Evensen, 2003) with zero mean and standard deviation based on perturbation values applied also in the more tested and robust TOPAZ system (Sakov et al., 2012). For the 2-m temperature, the standard deviation is $3\,\mathrm{K}$, cloud cover is $20\,\%$, per-area precipitation flux is $4 \times 10^{-9}\,\mathrm{m}$, and for wind, $1\,\mathrm{m\,s^{-1}}$ in both horizontal directions is applied. To account for model uncertainty the ice strength parameter, $P$, is perturbed. This is done by perturbing





the model parameter $C_f$ which is the frictional energy dissipation parameter. In CICE, $C_f$ is proportional to the ice strength (Hunke et al., 2015b),

$$P \propto C_f. \tag{4}$$

The default value of $C_f$ is 17, but according to Flato and Hibler (1995) this is not a well-known parameter. In our study, this parameter is modelled as a stochastic variable with a mean value of 17 and a standard deviation of 10, the different values are chosen based on values found during model tuning using observations by Flato and Hibler (1995). Since only one value less than 10 was found in their study, values less than 10 for $C_f$ is not used.

## 4.3 Experimental design

The assimilation model system consists of 20 ensemble members, with an assimilation time step of seven days. Similar to Sakov et al. (2012) a localization radius of 300 km is used. The initial ensemble is generated from ice states from the 1. January from 20 different years, while all ocean states are from the initial date 1. January 2010. Before performing the experiments, a model system assimilating ice concentration and sea-surface temperature (SST) from OSTIA (Donlon et al., 2012) is run for one year until 1. January 2011, to use as an initial state.

In the CICE model, the variables are distributed in 5 thickness categories, while all observations are single category values. This discrepancy was solved by assimilating the aggregated category values and using the EnKF correlation properties to update each category individually. After assimilations, the analysis results are post-processed before running new forecasts. During post-processing, it is verified that the consistency of the different ice variables is maintained during assimilation as the analysis can lead to for instance situations where some areas have a positive SIC but no volume, in this example the SIC is set to zero. In addition, all variable bounds are checked during post-processing. Due to linear correlation effects of the EnKF, locations with non-physical concentrations can occur, for instance, SIC values both above one and below zero.

For the ocean parameters, only ocean temperature and ocean salinity are updated during the assimilation. Experiments have shown that large instantaneous changes to the ocean parameters lead to model instability. These large changes are especially seen in the marginal ice zone (MIZ) where the ensemble spread is the largest and the update to the ensemble is the strongest. To prevent these instabilities in the ocean the magnitude of the ocean update during an assimilation step is limited. In this work, a maximum temperature update step of 0.2 K for the ocean surface layer and 0.1 K for all other ocean layers is chosen. Similarly for the salinity 0.2 for the surface layer and 0.1 for all other layers is chosen. The limits are chosen crudely, based on values where the model did not immediately crash after assimilation. Although this is a crude simplification, almost omitting ocean update, it is believed to be sound, because the focus in this research is on the sea ice, and because it is implemented consistently for all the model experiments.

Five assimilation experiments assimilating different observations are used to investigate the observations effect on the model. In experiment 1 only OSISAF SIC is assimilated, in exp. 2 both OSISAF SIC and CryoSat SIT, in exp. 3 both OSISAF SIC and SMOS SIT, in exp 4. OSISAF SIC and snow depth (SD) observations, exp 5. is a control run without assimilation. All





**Table 1.** Overview of the 5 experiments used to asses observation impact, the X marks if the given observation is assimilated in the experiment.

|  | OSISAF | Cryosat | SMOS | Snow depth |
|---|---|---|---|---|
| Exp1 (SIC) | X |  |  |  |
| Exp2 (SIC + SIT$_I$) | X | X |  |  |
| Exp3 (SIC + SIT$_R$) | X |  | X |  |
| Exp4 (SIC + SD) | X |  |  | X |
| Exp5 (Control) |  |  |  |  |

assimilation systems are initialized after one year of initial assimilation on 1. January 2011 and run for three years. A summary of the different experiments is shown in table 1.

## 5 Results

In this section, the output of the five ensemble experiments is compared. All results are based on the output from 2011-2013.
As mentioned, the first year of modelling, 2010, is only used to spin-up the experiments, generating a stable and consistent ice-ocean model state.

Many of the results shown in this section will be based on the root mean squared error (RMSE). In this study, the RMSE is weighted by the observation uncertainty, $\sigma_{Obs(i)}$,

$$RMSE = \sqrt{\frac{\sum_{i=1}^{N} \frac{(Mod(i) - Obs(i))^2}{\sigma_{Obs(i)}^2}}{N}}, \tag{5}$$

where N is the number of grid cells, Obs and Mod are the observations and model values, respectively. Thus an RMSE of one indicates that the difference between model and observations are on average of the same order as the observation uncertainty.

### 5.1 Validation against concentration observations

In Figure 2 the monthly averaged ensemble mean of the five experiments validated against two different SIC observation products, one assimilated and one independent is plotted. In Fig 2a the RMSE values of the ensemble mean of the modelled SIC validated with the observed AMSR-E/2 product are plotted after assimilation. All four assimilation experiments are found to be significant improvements compared to the control experiment without assimilation. Using a one-sided paired sample student t-test over all 36 months of simulation both the Cryosat internal SIT and SMOS rim SIT experiments show significant improvements compared to the OSISAF SIC only experiment on a 5 % level, but the differences are relatively small. The significance is derived using monthly data, but not yearly averaged as in the figures. However, the snow experiment is not found to be significantly better than the OSISAF SIC only experiment on a 5 % level, a p-value of 0.23 is found. The difference between the SIT experiments and the SIC only experiment is largest during the first half of the year, while in the second half of





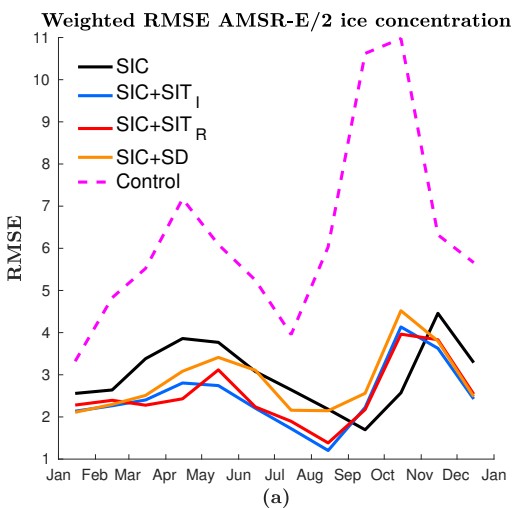
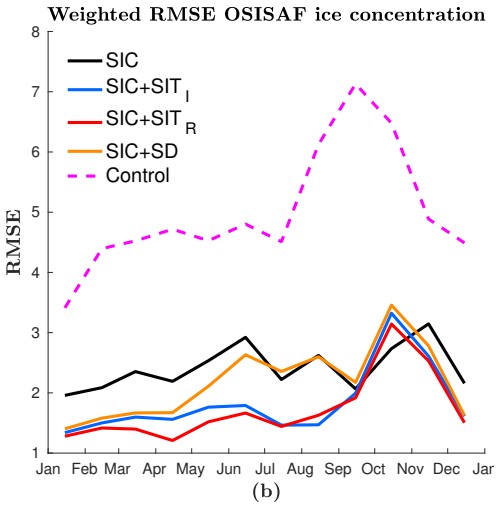

**Figure 2.** The monthly averaged RMSE of the ensemble mean SIC over three years. In a) the model is validated against AMSR-E/2 SIC observations and in b) OSISAF SIC observations. The lines represent different experiments, black: only SIC assimilation, blue: SIC and CryoSat thick internal SIT assimilation, red: SIC and SMOS and thin rim SIT assimilation, yellow: SIC and snow depth (SD) assimilation, magenta dotted: without assimilation.

the year all experiments give similar results with a peak in the RMSE in October-November. This peak in RMSE is also seen in the control model, indicating a possible model problem related to the transition from melt season to growing season.

In Fig. 2b the monthly averaged RMSE of the model SIC ensemble mean versus the assimilated OSISAF SIC observations is plotted. The result in Fig. 2b is similar to that of 2a, but the differences between the models are larger when verified against

OSISAF, even though the OSISAF observations are assimilated in all experiments. This is partly related to lower observation error in the MIZ for the OSISAF dataset than the AMSR-E/2 dataset, and that the OSISAF includes almost an extra year of observations, due to the AMSR-E/2 gap. Since the RMSE values are weighted by the observation error the differences in the MIZ are more pronounced when verified against OSISAF SIC observations. In addition, evidence that there are small differences between the two observation products is seen by different shapes on the graphs, even though the curves follow

the same trends. As mentioned, the Cryosat and SMOS SIT experiments are significantly better than the OSISAF SIC only experiment. When compared to the OSISAF observations also snow depth assimilation experiment is found to be significantly better than the OSISAF SIC only experiment, especially during the first half of the year there are significant differences. In conclusion, assimilating SIT and to some degree, snow depth has a significant effect on the SIC RMSE, and the effect is largest for the first half of the year. In the transition from melting ice to freezing ice, all four experiments give similar high RMSE

values.

RMSE estimates are sensitive to individual measurements contributing to large portions of the total RMSE, thus a small area with large errors will obscure the overall model results. Another assimilation quality measure is hit rates, where all grid cells are given equal weight in the analysis. In our work, the hit rate is analysed by classifying the SIC in three categories, open



water (concentration less than 10 %), low concentration ( < 50 %), and high concentration ( > 50 %). The hit rate is defined as the number of grid cells correctly classified. The independent AMSR-E/2 observations are used for verification. In Fig. 3a the number of grid cells correctly classified is shown; in Fig. 3b the number of grid cells with modelled ice and observed water; in Fig. 3c the number of grid cells with modelled water and observed ice; in Fig. 3d the number of grid cells with wrong concentration category, high SIC classified as low SIC and vice versa. All assimilation experiments outperform the control run in terms of hit rate. The control run has a large number of false positives, indicating too much ice. Among the experiments, the variations are small in spring, fall and winter, while summer shows significant differences. In summer all experiments have a minimum, this minimum is related to an under-prediction of sea ice and wrong classification of concentration in observations due to melt ponds on ice which leads to an underestimation of SIC in the observations (Kern et al., 2016). In summer the CryoSat assimilation has the highest number of hits, closely followed by the SMOS and snow experiments.

## 5.2 Total extent and volume

In figure 4, the total sea ice extent (4a), the total sea-ice volume (4b), and the total sea-ice volume overlapping the area and period covered by the CryoSat internal SIT observations (4c) are shown for the five experiments. Figure 4a shows that the control experiment has a too large sea-ice extent both in summer and winter, while the assimilation experiments have a slightly too large ice extent in winter.

The total sea-ice volume shown in Fig. 4b indicates large differences between the five experiments. Snow depth assimilation generally leads to thicker ice. The model has a lower amount of snow than the observations and due to positive correlation, the ice thickness is also increased during the assimilation of snow depth. The increased thickness can be seen by the fact that the snow depth experiment has about the same extent as the other experiments, but show a significantly larger ice volume, both in summer and winter for all three years. Both the SMOS and CryoSat ice thickness experiments lead to thinner sea ice compared to the control experiment. Especially the SMOS assimilation model shows much thinner sea-ice than do the other assimilation experiments. The thin SIT observations have a very strong effect on the modelled SIT, seen by an abrupt update of sea-ice volume during assimilation in winter. A concerning effect of the assimilation experiments is the strong decrease in the Arctic sea-ice volume throughout the period of study. The sea-ice volume maximum in winter is decreasing for every year of assimilation, this is not seen in the control run.

In Fig. 4c the modelled sea-ice volume is compared to the sea-ice volume in the combined CryoSat-SMOS product. The control model is found to have too thick ice compared to the observations, while the experiments assimilating SIT are much closer to the observations, though largely biased. It was seen that the decrease in sea-ice volume in Fig. 4b can be explained by the volume adjusting to the observed SIT. For the OSISAF SIC only assimilation experiment, the volume is also slowly diverging towards the observed volume, even though SIT is not assimilated. This is likely related to a more accurate sea-ice extent also leads to improved ice thickness in the marginal ice zone. However, the improvements are obtained at a slower pace than when assimilating SIT directly.





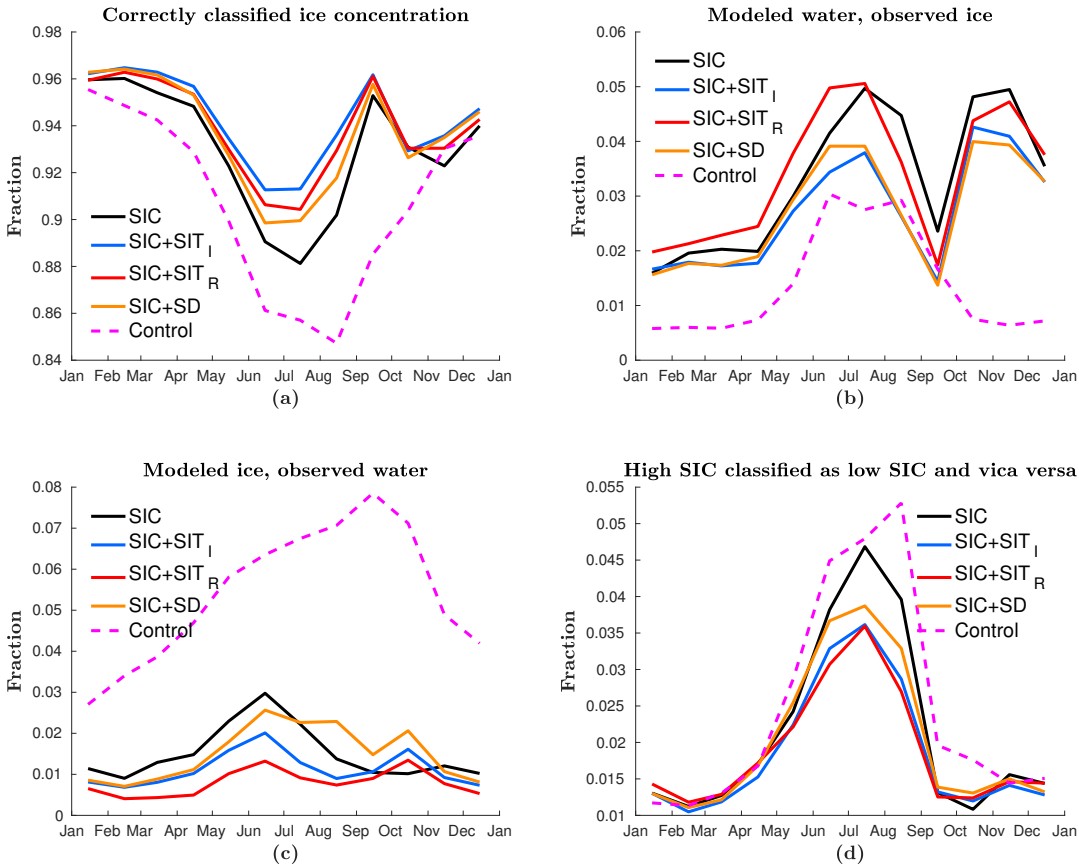

**Figure 3.** Classification of the model result based on three classes, high concentration ice ($> 50\,\%$), low concentration ice ($> 50\,\%$) or water and compared to AMSR-E/2 SIC observations. The figures show a) the fraction of correctly classified grid cells, b) the fraction of grid cells with modelled ice while water is observed, c) fraction of grid cells with modelled water while ice is observed, and d) fraction of grid cells where the model and observations have different SIC classification. The colour coding in the figure is the same as that of Fig. 2. These panels cover all possible classifications, thus the sum of them equals to one.



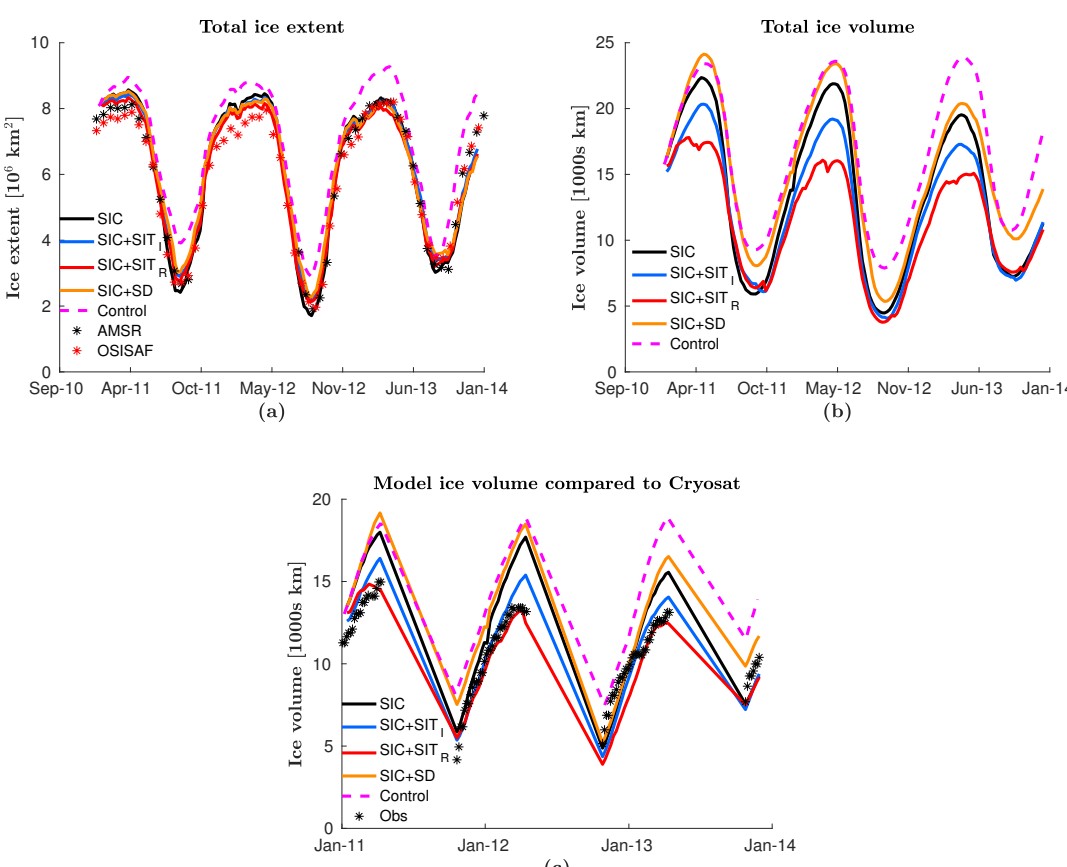

**Figure 4.** The evolution of a) total sea-ice extent, b) total sea-ice volume, and c) total sea-ice volume for the area covered by the Cryosat-SMOS SIT observation product. The coloured lines represent the same as in Fig. 2. In a) the black stars represent the AMSR-E/2 SIC observations and the blue stars the OSISAF SIC observations. In b) same as a) without observations. In c) the black stars represent the Cryosat-SMOS observation product.

## 5.3 Validation against thickness observations

In Fig. 5a the SIT RMSE of the ensemble averaged modelled SIT is verified with the combined SMOS-CryoSat SIT product. The experiment assimilating SMOS thin SIT has significantly lower RMSE values than the other three assimilation experiments. The other three experiments are more similar, all showing high RMSE values. It is found using a one-sided paired student t-test that only the SMOS SIT experiments are significantly better than the SIC only assimilation, with p-values less than 5 %. Due to the high RMSE values, only small improvements compared to the control run is seen. The result is consistent with what was found for the sea-ice volume (Fig. 4c), regarding the SMOS SIT assimilation having the strongest effect on the modelled SIT. The reason for the high RMSE values is that in general, the model has to much ice, leading to too thick ice in the MIZ. For the SMOS-CryoSat SIT product the uncertainties provided are very small, especially in the MIZ where the SMOS





observations are used, thus when calculating the RMSE these values have a huge effect on the result. Thus it is also reasonable that the assimilation system assimilating these MIZ thickness observations also give the lowest RMSE values. For the other assimilation systems, the ice extent is updated in the MIZ, but the thickness reduction takes longer because this has to evolve over time.

As for the SIC observations, the RMSE values are biased by locations showing large differences. Particularly for thickness which is not bounded upwards, a few grid cells in the MIZ can contribute to a large total RMSE. As for concentration, an alternative measure where correctly classified model thickness hit rates are used. The model is separated into four thickness categories, less than $0.5\,\mathrm{m}$, between $0.5\,\mathrm{m}$ and $1.5\,\mathrm{m}$, between $1.5\,\mathrm{m}$ and $3\,\mathrm{m}$, and above $3\,\mathrm{m}$. In Fig. 6a the number of correctly classified ice thicknesses grid cells is plotted for each experiment. The figure shows that the CryoSat internal SIT experiment

is the model which has the highest number of correctly classified grid cells. The other experiments are more similar, except in spring where the SMOS rim SIT assimilation is equally good as the CryoSat internal SIT assimilation, and both much better than the other three. In spring the SIC only and snow depth assimilations are not improved compared to the control case. In general, the model shows too much ice. This can be seen by a large number of grid cells having too thick ice in the control model (Fig. 6b). This is a combination of the sea-ice extent being too large and the ice is too thick. By assimilating observations

the ice volume is reduced, not only for the SIT assimilations, but also for the snow depth and SIC only assimilations, but to a lower degree for the latter. This is an effect of a lower sea-ice extent (Fig. 4a). In Fig. 6c the number of grid cells with too thin ice compared to the observations is shown. It was found that this problem is large in early winter for all experiments but reduces during winter for all experiments except the SMOS experiment. During SMOS assimilation, only thin ice is assimilated which might lead to a bias towards the thinner ice, causing a relatively high number of grid cells with too thin ice.

As an example, in Fig. 7 the absolute differences between the experiments and the combined CryoSat SMOS ice thickness observations are plotted for 1. April 2011. Figure 7 is consistent with Fig. 6a showing that the CryoSat experiment has the smallest differences compared to the observations in the internal Arctic, affecting a large area, however, large differences can be seen in MIZ. While for the SMOS rim SIT assimilation the effect is the opposite, with large impact in the MIZ, and low impact in the ice interior. This shows that assimilating SIT is important both in the interior and in the MIZ.

In addition to the satellite observations, the independent airborne IceBridge dataset is used for verification of the modelled SIT (Kurtz et al., 2013, 2014a). This dataset has low temporal and spatial distribution, but is believed to have higher accuracy and has a much higher spatial and temporal resolution. All observations occurring in March and April are gathered as a yearly averaged observation as a function of space. These yearly observations are then compared to modelled SIT averaged over the same period for the observed IceBridge locations. Since the IceBridge resolution is much higher than that of the

model, all IceBridge observations within one model grid cell are averaged and used for verification, the average is done by a weighted mean based on the observation uncertainty. The validation results are shown in table 2. On average the CryoSat SIT experiment has the best SIT estimation as compared to IceBridge. Both the SMOS and the CryoSat SIT experiments give on average thinner SIT than the IceBridge observations, consistent with the findings of Chen et al. (2017). The last line in the table shows the RMSE between the CryoSat observations and the IceBridge observations and the results show that the error is

comparable to the model errors.





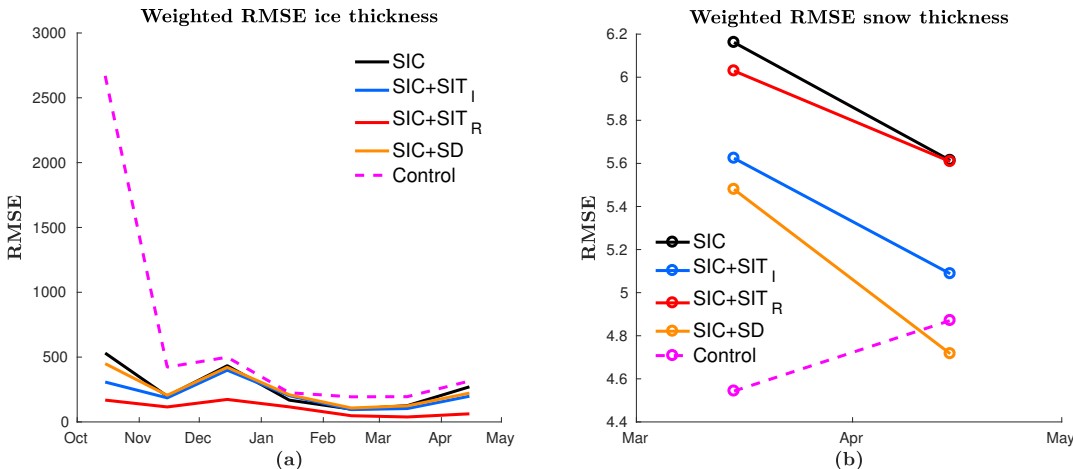

**Figure 5.** RMSE of monthly averaged model SIT and snow depth averaged over all ensemble members calculated against observed a) ice thickness and b) snow depth. The colour coding is as in Fig. 2.

**Table 2.** The yearly averaged RMSE values of the five experiments compared to the IceBridge aerial SIT observations. Bold values represent the model with the lowest RMSE values for that year.

|                | 2011 | 2012 | 2013 |
|----------------|------|------|------|
| SIC            | 0.88 | 0.87 | 1.11 |
| SIC+SIT$_I$    | **0.63** | **0.86** | **0.72** |
| SIC+SIT$_R$    | 0.74 | 1.14 | 0.87 |
| SIC+SD         | 0.93 | 1.38 | 1.64 |
| Control        | 0.82 | 1.25 | 2.31 |
| Cryo Obs       | 0.67 | 0.95 | 0.84 |

For all three years, the CryoSat assimilation has lower RMSE values than the CryoSat observations, indicating a well-balanced assimilation, with appropriate observation error and ensemble spread. It should also be mentioned that the Cryosat observations have less spatial coverage than the model and not all IceBridge observations are covered, thus the number of useful observations for the CryoSat RMSE calculation is smaller than for the validation of the experiments.

## 5.4 Validation against Snow observations

In Fig. 5b the RMSE of monthly averaged modelled snow depth over all ensembles validated against the observed satellite snow depth is plotted. The control experiment is found to have the lowest RMSE values. This is most likely an effect of sea-ice extent being different compared to the assimilation experiments, rather than the assimilation decline the accuracy of the modelled snow depth. In addition, the control experiment has an increasing RMSE during the period, while the assimilation experiments show the effect of assimilation by decreasing the RMSE. For the assimilation experiments, the snow experiment



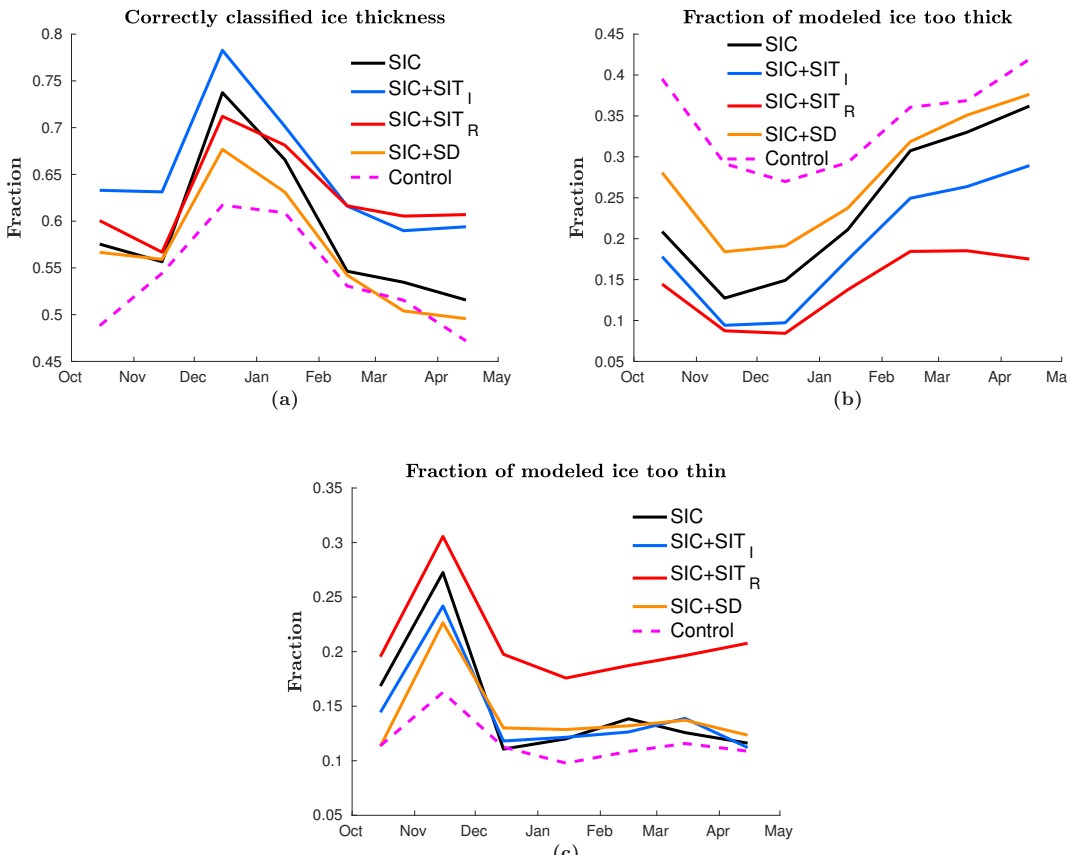

**Figure 6.** The monthly mean SIT averaged over all ensemble members is classified into four thickness categories and compared to the CryoSat-SMOS SIT observation product. The fraction of grid cells are shown with a) correctly classified thickness category, b) too thick ice, and c) too thin ice. As in Fig. 2 the coloured lines represent different experiments.

has the lowest RMSE values followed by the CryoSat experiment, indicating that the thick ice observations have a correlation effect on the snow depth. These two observation products also cover a similar area of the Arctic ocean.

A verification of the modelled snow depth using the independent IceBridge dataset is given in table 3. The same method as for the SIT in table 2 was used. It is found that none of the experiments is particularly better than any of others when verified against IceBridge snow depth observations. This lack of improvement can be an indication of a too simple snow component in our coupled system, only one snow layer is used. It was also seen that within one grid cell there where huge variations in the IceBridge snow observations, indicating the difficulty of modelling snow with a coarse resolution model. In addition, the snow observations are in the early development stage and might have larger uncertainties than what used in this study.





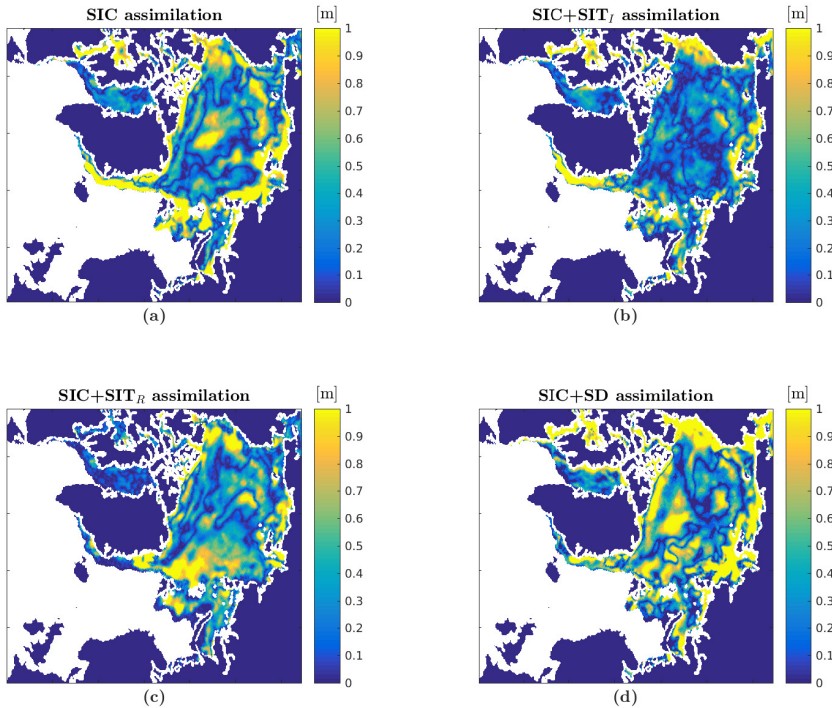

**Figure 7.** Absolute differences between the experiments and the combined SMOS-CryoSat observation product are given on 1. April 2011. The experiments are assimilating a) OSISAF SIC, b) OSISAF SIC and CryoSat SIT, c) OSISAF SIC and SMOS SIT and d) OSISAF SIC and snow depth.

**Table 3.** The yearly averaged RMSE values of an ensemble average of snow depth averaged over all the ensemble members for the five experiments compared to the IceBridge airborne snow depth observations. Bold values represent the model with the lowest RMSE values for that year.

|  | 2011 | 2012 | 2013 |
|---|---|---|---|
| SIC | 0.79 | 1.38 | 2.64 |
| SIC+SIT$_I$ | 0.79 | 1.15 | **1.44** |
| SIC+SIT$_R$ | 0.78 | **0.83** | 1.73 |
| SIC+SD | **0.74** | 1.22 | 1.46 |
| Control | 0.77 | 2.49 | 1.85 |
| Snow Obs | 1.46 | NA | 1.17 |



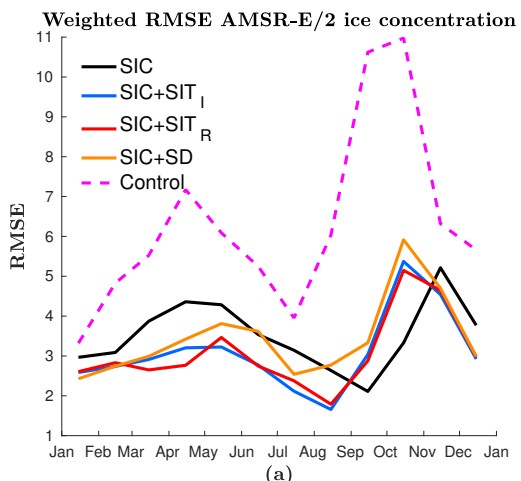
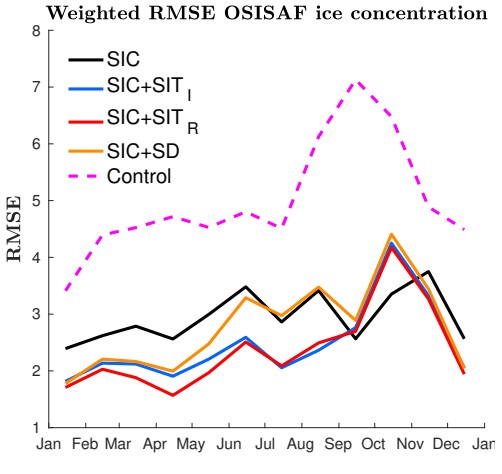

**Figure 8.** RMSE of a seven day monthly averaged ensemble average of modelled SIC forecast validated by a) AMSR-E/2 SIC observations and b) OSISAF SIC observations.

## 5.5 One week forecasts

Figure 8 shows the RMSE of the monthly averaged modelled SIC over all ensemble members before assimilation validated against the AMSR-E/2 and OSISAF SIC observations. Since the assimilation time step is seven days, this gives the accuracy of a seven-day forecast. The comparison against AMSR-E/2 SIC observations (Fig. 8a) shows that the differences between the experiments are small, and the differences are similar to those found after assimilation (Fig. 2a). In general, the system with the most accurate initial state also gives the most accurate forecasts. Thus the CryoSat and SMOS SIT assimilation experiments have a better seven-day forecast from January to June than SIC only, and snow depth assimilation shows improvements from January to April. Using the OSISAF SIC observations (Fig. 8b) gave the same result as found for AMSR-E/2: the best initial states also provide the best forecast, indicating that the sea-ice overall does not change much in a week. The same experiments were also done for ice thickness and snow depth and similar results were encountered. A reason for the small differences between the different experiments is the coarse model resolution. Large-scale variations as seen by a 20 km model are not expected within a week.

## 6 Seasonal forecast

In the previous section, it was found that our coarse resolution model only exhibits small changes during a one week forecast. Thus a more interesting forecast would be one of seasonal length. A five-month forecast of the September sea-ice extent is performed. This is done by running each of the experiments from April to September without assimilation and validating against the OSISAF SIC observations. In Fig. 9a, the RMSE of three five-month forecasts are shown sequentially, and a monthly averaged RMSE over the three years is shown in Fig. 9b. The figures show that that the experiments have very similar





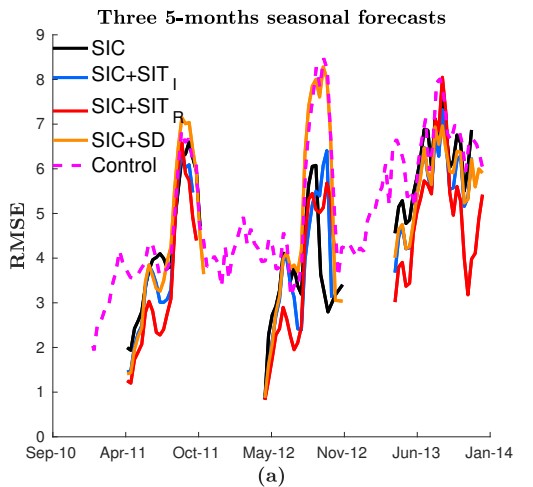
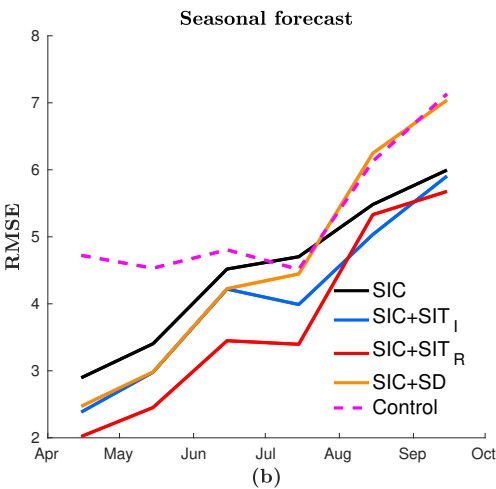

**Figure 9.** Seasonal forecast of summer sea-ice extent. Each forecast started at the beginning of April every year. The figures show SIC RMSE averaged over all ensemble members. The coloured lines represent the same as in figure 2. In a) Full period, in b) monthly averaged values.

seasonal forecasts, with some differences in late summer. In general, the model error is gradually increasing towards the level of the control run, and in summer they have similar error levels. In August/September the experiments assimilating thickness and concentration seems to be improvements compared to without assimilation and assimilating snow depth observations. All experiments show an increased RMSE in 2013, this is related to a too low sea-ice extent. The low sea-ice extent is caused by a

weaker modelled ice growth compared to observations in the first months of 2013.

The seasonal forecast is compared to a climatological seasonal forecast in Fig. 10. This provides an estimate of the expected sea-ice forecast accuracy. The climatological experiment is done by running the model with averaged atmospheric forcing data over the years from 2000 to 2014. The result shows that the forecast skill of the model is rapidly decreasing and that a correct atmospheric forecast is very important for an accurate sea-ice forecast. But still, skills are evident on much longer time-scales

that can be obtained with numerical weather prediction models.

## 7   Discussion

Significant differences in modelled SIC after assimilation was found, especially in the first half of the year. The SMOS and CryoSat SIT assimilations gave the lowest RMSE values, significantly better than assimilating OSISAF SIC only. The snow depth experiment showed improvements during the first half of the year compared to the experiment assimilating OSISAF

SIC observations only. In addition, assimilating SIT and snow depth lead to an improved model of SIC in summer, where the CryoSat internal SIT assimilation gave the highest number of correctly classified grid cells, closely followed by the SMOS rim SIT and snow depth assimilations. The reason for these differences in summer is that the pace in which the ocean becomes ice-free is dependent on the ice thickness and the snow depth. In the second half of the year, fall and early winter, all our experiments gave similar results, these similarities seem to be a consequence of the transition from melt season to growing





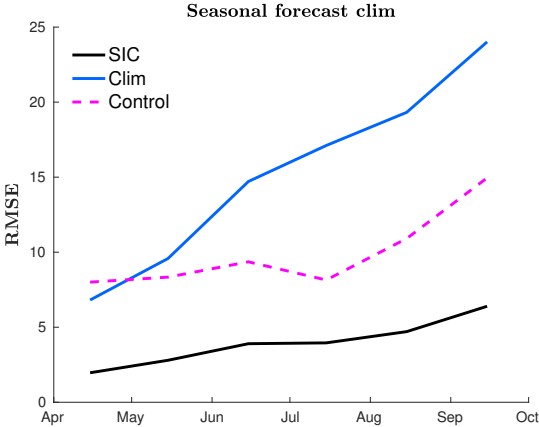

**Figure 10.** Seasonal forecast of summer sea-ice extent both with climatological forcing and re-analysed forcing. Each forecast started at the beginning of April every year. The figure describes monthly averaged RMSE SIC averaged over all ensemble members. The blue line represents a forecast using a climatological forcing made from an over of atmospheric data for 2000-2014 with assimilation, the black line using re-analysed atmospheric forcing and assimilation, and the dotted magenta line use re-analysed forcing only.

season not well represented in the model. The observed transition is faster than the modelled, leading to an extended period with more open water in the model than in the observations.

In the control model without assimilation, the ice extent both in summer and winter was found to be larger than observed. However, with assimilation, the experiments are closer to the observed extent, even though a slight overestimation of extent
in winter was found for the first two years. The sea-ice extent overestimation in winter is a result of a lower effect of SIC assimilation in winter due to lower ensemble spread. When the ensemble spread is low the EnKF assimilation result is closer to the model values, because the estimated model errors become small.

It is found that originally the sea-ice volume is too large compared to the observations, and over the three years, the sea-ice volume in the assimilation experiments are gradually decreasing towards the observed values in the SMOS-CryoSat SIT
product. The effect is much stronger for the SMOS rim SIT assimilation, indicating that a large portion of the original sea-ice volume overestimation is located in the MIZ. This is a consequence of too much ice in the control model causing the observed MIZ to be located deeper into the Arctic as compared to the model, as noted by Fritzner et al. (2018).

In the verification of modelled SIT (Fig. 5a), the SMOS rim SIT assimilation was found to give the lowest RMSE values, while the CryoSat internal SIT assimilation had the largest amount of correctly classified thickness grid cells. This is as
expected since even though the CryoSat observations cover a larger area, they are 30-day averaged observations with much larger uncertainties than the SMOS observations. In addition, the non-updated grid cells in the MIZ leads to larger RMSE values than non-updated grid cells in the internal Arctic, where the model, in general, is more accurate and less sensitive to changes. When verified by IceBridge observations which only covers the central Arctic, the CryoSat SIT assimilation experiment was found to give the lowest SIT RMSE values. The CryoSat SIT observations are in general thinner than the SIT values for the
SIC only experiment. In comparison with the IceBridge observations, the CryoSat SIT is biased low, which was also found by



Chen et al. (2017). When assimilating snow depth, it was found that snow depth observations, in general, were thicker than those modelled, resulting in increased snow depth during assimilation. Due to the correlation nature of the EnKF, a positive correlation between snow depth and SIT resulted in increased SIT in the snow depth assimilation experiment compared to the other assimilation experiments.

Validating our experiments with snow observations proved the control run to have the lowest RMSE values, this can be an effect of a different sea-ice extent in the control run than in the assimilation experiments. For the control model, the ice extent is too large, thus collecting more snow on the ice than the assimilation experiments. When the ice concentration is reduced during assimilation, the accumulated snow is also removed, this can result in removal of too much snow if the ice extent is less than it should be. A verification of the impact of assimilation on the snow depth is that the RMSE is decreasing throughout the

observation period for the assimilation experiments, while for the control run the RMSE is increasing. Between the assimilation experiments, the snow depth assimilation was found to give the lowest snow depth RMSE values, not unexpected since the same dataset is used for assimilation and verification. More interestingly the CryoSat internal SIT experiment has significantly lower RMSE values than the SMOS rim SIT and OSISAF SIC only assimilations, indicating a close correlation between SIT and snow depth. A curiosity here is that the SIT assimilation has a positive effect on the snow depth, while it was found previously

that the snow depth assimilation had a negative effect on the SIT. This is likely an effect of more SIT observations than snow depth observations, and SIT is assimilated throughout the whole winter. It could be the case that the correlation relationship between snow depth and SIT changes throughout the winter. This results in a better snow depth estimate, while for snow depth assimilation the assimilation period is limited to Mars-April. In addition, when the assimilation is only in two months of the year, the model error is larger when the assimilation period starts, thus the assimilation update has a large effect both on snow

depth and SIT. An indication of the relation between SIT and snow depth is also seen by lower snow depth RMSE value for the SMOS rim SIT assimilation than the OSISAF SIC only assimilation. Since the SMOS system covers a much smaller area and has less overlap with the snow dataset than the CryoSat internal SIT, the effect of assimilation on the modelled snow is smaller.

When validating our experiments with the IceBridge snow depth observations, none of the experiments showed any improvements compared to the others. This can be related to an underestimated uncertainty in the snow observations, or that the

snow representation in the model is too simplistic, only utilizing a single layer. Another problem is local variations, the coupled model is coarse with a resolution of 20 km, but as can be seen from the IceBridge observations the snow depths have large spatial variations in this range. This causes high RMSE values, both compared to satellite observations (on the model grid) and the modelled snow depth values. In addition only four months of snow depth observations where available for assimilation during the three years.

For sea ice, the model drift is in general small, the model system with the best initial state provides the best short-term forecast. The main parameters analysed in this study snow depth, SIT and SIC all vary on longer time scales than one week for the spatial resolution in our model. Thus the correlation between day-one and day-seven is too strong, as also shown by Chen et al. (2017) the sea-ice drift is low.

Several five-month seasonal forecasts of the September sea-ice extent showed small differences between the assimilation

experiments. All experiments showed a steady increase in RMSE with time. This is likely caused by the model overgrowth of





ice. The seasonal forecasts showed that after 3-4 months the RMSE values were found to be of the same order as those in the control run. Thus assimilation gives at least an improvement over 3-4 months, and the September result suggests that with the assimilation of SIC and SIT there are improvements in the Arctic sea-ice extent compared to the control run on even longer timescales, this was not seen for the snow depth experiment. The seasonal forecast was compared to a climatological run, and

it was found that without accurate forcing the forecast degenerates fast.

In this work, four different observation products have been used for assimilation. The different products differ widely in both temporal and spatial coverage in addition to accuracy. There is no doubt that it is preferable to have as much coverage and as accurate observations as possible. Where a realistic observation error is a necessity for the assimilation, without a realistic observation error the observation is not useful. E.g. the Cryosat product does not provide an observation error and a uniform

error was chosen, which will lead to some observations given too much weight and others with too little. In this study, the spatial resolution of the observations is not a problem, because the model resolution is coarse, but in the future when the model resolution increases, there will be an increasing demand for high-resolution SIT observations. Both SIT products are only available in winter, and temporal coverage of the snow depth observations are limited to four months out of a three-year experiment. Thus for these products to be even more useful, there is a strong need for increased seasonal coverage, especially

in summer when the Arctic sea-ice extent is at a minimum and there is ship traffic there. Observations can then help to improve the models, thereby helping planning operations and decrease the risks. In addition, because there are few snow observations available for assimilation, there are large unknown aspects regarding the assimilation effect. Finally, it should be remembered that the model itself has in general too much ice and that the forcing is known to contain biases and errors (Jakobson et al., 2012).

**8   Conclusions**

In conclusion, we have found that assimilation of more observation types than SIC into coupled sea-ice ocean models can lead to significant model improvements. We show that especially the assimilation of SIT leads to improvements in modelled SIC, SIT and snow depth, for long-term model improvement. There is a clear evidence that assimilation of SIT gives a better representation of the full ice state and we recommend that they are assimilated into models when available. Even though

SIT seems to be an important variable for improving sea-ice modelling, it still has several limitations in terms of spatial and temporal resolution and realistic observation errors.

Assimilation of snow depth was found to have a weaker effect on the model than assimilating SIT, but improvements to modelled SIC and modelled snow depth were found. In addition, we found a strong correlation between SIT and snow depth which should be analysed further when more observations from other months become available. The low efficiency of snow

depth observations can be related to a too simple snow component with only a single layer used or low model resolution. It is also important to keep in mind that the snow depth observations are in an early development stage, and we should expect improvements in the future. Possibly inaccurate observations or a wrong uncertainty estimation can have a huge impact on





the assimilation result. Due to the small temporal coverage in our study more investigation has to be done on the effect of assimilating snow depth observations.

As mentioned the assimilation of SIT leads to an improved model, which leads to improved short-term forecasts over time, because the initial states are better represented. For seasonal forecast, we found that the model improvements due to assimilating observations have a memory of at least 3-4 months, and possibly even longer. Assimilating SIC and SIT showed improvements of the September ice forecasts compared to assimilating snow depth and no assimilation.

*Competing interests.* No competing interests.

*Acknowledgements.* We would like to thank Pavel Sakov for help using and implementing the EnKF-c code and for helpful discussions regarding the EnKF. We thank the EUMETSAT OSISAF centre and the NASA National Snow and Ice Data Center for providing the sea-ice concentration data, and the Integrated Climate Data Center at the University of Hamburg for the SMOS ice thickness observations. We would like to thank NASA National Snow and Ice Data Center for providing the Cryosat sea-ice thickness observation, and the Icebridge snow and sea-ice thickness observations.

This work was funded through the Center for Integrated Remote sensing and Forecast for Arctic Operations through the Norwegian Research Council grant no. 237906. Rostosky was supported by the Transregional Collaborative Research Center (TR 172) "ArctiC Ampli-fication: Climate Relevant Atmospheric and SurfaCe Processes, and Feedback Mechanisms (AC)[3], which is funded by the German Research Foundation (DFG). Two supercomputers provided by the Norwegian Metacenter for Computational Science (NOTUR) was used for the computational work, the Vilje computer at the Norwegian University of Science and Technology was used with project and the Stallo computer at the University of Tromsø both under project NN9348K.

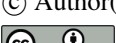


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
