# Peer review of "Impact of assimilating sea ice concentration, sea ice thickness and snow depth in a coupled ocean-sea ice modeling system"

_The Cryosphere, 2018_

## Referee Comment (RC1) · Anonymous Referee #1 · 16 Oct 2018

Review on "Impact of assimilating sea ice concentration, sea ice thickness and snow depth in a coupled ocean-sea ice modeling system", by Fritzner et al., submitted for publication in The Cryosphere Discussion.

General comments :

The paper shows the results of sea ice data assimilation experiments into a coupled ocean and sea ice model using an ensemble Kalman filter (EnKF). Sea ice concentration, sea ice thickness and snow depth are assimilated in different combinations and verifications are performed using assimilated and independent observations. The impact of assimilation is measured on the analysis, 7-day forecasts and 5-month seasonal forecasts. The paper is generally well written. The assimilation experiments and verifications are well designed. The assimilation of snow depth is particularly original as it has not been done in other studies, as far as I know. However, some aspects could be clarified. For examples, the observation-error used for the assimilation could have been explicitly specified. In some cases it is not clear whether the verification has been done on the ensemble mean or on individual ensemble members (and then calculating and average). Also the error of the ensemble mean and ensemble spread relationship could have been shown, as this is usually considered a requirement for an EnKF.

Specific comments :

1. A couple of sentences at the end of the abstract around line 15 are confusing to me. It seems that the conclusions about the assimilation of snow depth are contradictory: "... while the snow observations have a positive effect on snow thickness and ice concentration. In our study, the seasonal forecast showed that assimilating snow depth lead to a worse estimation of sea-ice extent compared to the other assimilation systems, the other three gave similar results." How come the assimilation of snow depth have a positive effect on ice concentration but lead to a worse estimation of sea-ice extent ?

2. In section 4.1, page 7: First Pb is defined as the background-error covariance matrix. A couple of lines later, it is referred as the model-error covariance matrix. I think you should stick to background-error covariance matrix because model-error covariance matrix is usually reserved for the errors accumulated during model integration.

3. In section 4.3, it is mentioned that there is 5 thickness categories; I assume they are the partial concentrations for each thickness categories and that the total ice concentration can be calculated from them. Later it is mentioned that the assimilation can result into a positive SIC but no volume. Does that mean that the 5 thickness categories and the SIC are all independent analysis variables ? If that is the case, wouldn't

it be better to only have the 5 partial concentrations as analysis variable and calculate SIC ? It seems that it would avoid the problem of having positive SIC but no volume.

4. In Figure2, what are the observation uncertainty of AMSR-E/2 ice concentration used in the calculation of the RMSE ? Are they included in the product and are they constant values or are they specified for each points ?

5. Figure 5: Over which year(s) ? Is it against IceBridge observations ? Also in the text under section 5.4, please specify what is "observed satellite snow depth", is this IceBridge ?

6. Page 16, line 5: "This lack of improvement can be an indication of a too simple snow component in our coupled system, only one snow layer is used." I think that is pure speculation, unless the authors can show evidence to convince the readers. Could the reason be simply that there are large discrepancies between IceBridge and the assimilated snow depth products ? Same comment on page 21, line 25 and on page 22, line 30.

7. Table 3: It is hard to understand from the caption what are the numbers in the table. Is the averaged snow depth over a grid cell compared to the model ensemble mean ? It would help rephrase the caption, maybe removing one of the 3 averages and using "ensemble mean", if that is appropriate.

8. Figure 8: Are these monthly averages over the 3 years ? The caption is hard to read because the "seven day" and "forecast" are too far apart.

9. Figure 9: Please specify in the caption that (b) is the monthly averaged RMSE over the three years.

10. Figure 10: I think it is unrealistic to use re-analysed forcing for the seasonal forecasts, as the re-analysed forcing would not be available in an operational real time context.

Technical corrections :

Page 5, line 8: Change "…observations where given..." to "…observations were given..."

Caption of Figure 3 : Change "low concentration ice (> 50 %)" to "low concentration ice (< 50 %)"

Caption of Figure 4 : Change "the blue stars the OSISAF" to "the red stars the OSISAF".

Caption of Figure 4 : For ice volume the units are km. Is it the volume per unit area ?

Caption of Figure 4 : It would be easier for the readers to mention that the x-labels are month-year.

Page 14, line : Change "sea-ice extent being too large and the ice is too thick" to "sea-ice extent being too large and the ice being too thick".

Page 18, line 18: "The figures show that that …"

Figure 10: Change "The blue line represents a forecast using a climatological forcing made from an over of atmospheric data for 2000-2014 with assimilation" to "The blue line represents a forecast using a climatological forcing made from atmospheric data over 2000-2014 with assimilation"

Page 21, line 31: Change "The main parameters analysed in this study snow depth, SIT and SIC all vary on longer time scales than one week for the spatial resolution in our model" to "The main parameters analysed in this study, snow depth, SIT and SIC, all vary on longer time scales than one week for the spatial resolution in our model"

---

## Author Comment (AC1) · 13 Nov 2018

Review on "Impact of assimilating sea ice concentration, sea ice thickness and snow depth in a coupled ocean-sea ice modeling system", by Fritzner et al., submitted for publication in The Cryosphere Discussion.

General comments :

The paper shows the results of sea ice data assimilation experiments into a coupled ocean and sea ice model using an ensemble Kalman filter (EnKF). Sea ice concentration, sea ice thickness and snow depth are assimilated in different combinations and verifications are performed using assimilated and independent observations. The impact of assimilation is measured on the analysis, 7-day forecasts and 5-month seasonal forecasts. The paper is generally well written. The assimilation experiments and verifications are well designed. The assimilation of snow depth is particularly original as it has not been done in other studies, as far as I know. However, some aspects could be clarified.
We thank the reviewer for the kind words and the careful and constructive feedback.

For examples, the observation-error used for the assimilation could have been explicitly specified.
Information regarding the observation uncertainty for AMSR sea-ice concentration, SMOS sea-ice thickness and the snow depth product is now provided in the observations section.

In some cases it is not clear whether the verification has been done on the ensemble mean or on individual ensemble members (and then calculating and average).
Verification is always done on the ensemble mean. This has been clarified by writing ensemble mean instead of ensemble average when applicable.

Also the error of the ensemble mean and ensemble spread relationship could have been shown, as this is usually considered a requirement for an EnKF.
We agree that this would be an interesting result to show, but we think that for sea-ice this is less interesting than it likely is for other applications. We think it would be difficult to present in a useful manner in our case. In a normal EnKF system one would expect the ensemble spread to be of the same order as the ensemble mean error, but this is not the case here. To generate the ensemble spread we use perturbation amplitudes we find to be physically reasonable, we set this independent of model error. This creates an ensemble spread significantly lower than the model error. In addition, too large ensemble spread with only 20 ensemble members would give an assimilation result skewed towards the observations.

Specific comments :

1. A couple of sentences at the end of the abstract around line 15 are confusing to me. It seems that the conclusions about the assimilation of snow depth are contradictory: ". . . while the snow

observations have a positive effect on snow thickness and ice concentration. In our study, the seasonal forecast showed that assimilating snow depth lead to a worse estimation of sea-ice extent compared to the other assimilation systems, the other three gave similar results." How come the assimilation of snow depth have a positive effect on ice concentration but lead to a worse estimation of sea-ice extent ?

We agree that this is not well formulated. The point we make here is that there is a difference between long- and short-term effect of the assimilation. Where a positive effect is seen at the shorter timescales, immediately after assimilation and for the one-week forecasts. While for the seasonal forecast over several months there is a negative effect of assimilating the snow depth observations. The text has been updated for clarification: «It is found that the assimilation of ice thickness strongly improves ice concentration, ice thickness and snow depth, while the snow observations have a smaller but still positive short-term effect on snow thickness and ice concentration. In our study, the seasonal forecast showed that assimilating snow depth lead to a less accurate long-term estimation of sea-ice extent compared to the other assimilation systems, the other three gave similar results.»

2. In section 4.1, page 7: First Pb is defined as the background-error covariance matrix. A couple of lines later, it is referred as the model-error covariance matrix. I think you should stick to background-error covariance matrix because model-error covariance matrix is usually reserved for the errors accumulated during model integration.

This has been changed.

3. In section 4.3, it is mentioned that there is 5 thickness categories; I assume they are the partial concentrations for each thickness categories and that the total ice concentration can be calculated from them. Later it is mentioned that the assimilation can result into a positive SIC but no volume. Does that mean that the 5 thickness categories and the SIC are all independent analysis variables ?

The 5 partial SICs and the total SIC is 6 different parameters in the analysis, but they are not independent, the total SIC is only a sum of the 5 partial SICs. The model uses the 5 partial SICs, while the total SIC is a dummy parameter used for assimilation. When assimilating, the total SIC is the parameter corresponding to the observations, while the partial SICs are updated based on correlation and these are the ones used in the model afterwards.

If that is the case, wouldn't it be better to only have the 5 partial concentrations as analysis variable and calculate SIC ? It seems that it would avoid the problem of having positive SIC but no volume.

If we understand you correctly this is what is already done.  What we mean is that we can have a partial SIC larger than one, but the corresponding partial SIT zero or less than zero. New text: where some areas have a positive partial SIC but the corresponding partial SIT is zero or less than zero.

4. In Figure2, what are the observation uncertainty of AMSR-E/2 ice concentration used in the calculation of the RMSE ? Are they included in the product and are they constant values or are they specified for each points ?

The AMSR observation uncertainty is included in the product and specified in each grid point. This is now specified in section 3: Observations. New text: «The AMSR-E/2 SIC observation product includes individual uncertainty estimates for all grid points. This uncertainty is based on the sum of algorithm uncertainty and smearing uncertainty. Where smearing uncertainty is related to the location of the observation compared to the grid.»

5. Figure 5: Over which year(s) ? Is it against IceBridge observations ? Also in the text under section 5.4, please specify what is "observed satellite snow depth", is this IceBridge ?

No, this is against the observed snow depth product used for assimilation. New caption: «RMSE of monthly averaged model SIT and snow depth averaged over all ensemble members for the years 2011-2013 calculated against the a) combined SMOS-Cryosat SIT product and b) observed snow-depth product. These are observations also used for assimilation.»

New text in section 5.4: «In Fig. \ref{fig:Snow_Thickness}b the RMSE of monthly averaged modelled snow depth over all ensembles validated against the observed satellite snow depth \citep{Rostosky_2018} used for assimilation is plotted.»

6. Page 16, line 5: "This lack of improvement can be an indication of a too simple snow component in our coupled system, only one snow layer is used." I think that is pure speculation, unless the authors can show evidence to convince the readers. Could the reason be simply that there are large discrepancies between IceBridge and the assimilated snow depth products ? Same comment on page 21, line 25 and on page 22, line 30.
We agree that this is pure speculation and is mentioned as a suggestion to what might cause the lack of consistency between model and observations. As you mention there are large errors in the snow depth observation product too as compared to icebridge, and we agree this is a more likely reason for the large errors. We have changed the text to highlight this: «It is seen that within one grid cell, there are huge variations in the IceBridge snow observations. Such variations cannot be provided with a coarse resolution model. Hence large errors are found for the RMSE against IceBridge observations for experiments where the snow observation are assimilated, even though IceBridge is used to «tune» the assimilated product \citep{Rostosky_2018}. In addition, the snow component used in our coupled system is likely too simple, having only one snow layer, which may effect the snow cover accuracy.»

7. Table 3: It is hard to understand from the caption what are the numbers in the table. Is the averaged snow depth over a grid cell compared to the model ensemble mean ?
There was an error here, the ensemble mean is validated by observations averaged over all grid cells.

It would help rephrase the caption, maybe removing one of the 3 averages and using "ensemble mean", if that is appropriate.
We agree, new caption: «The annual-mean RMSE of the ensemble-mean snow depth averaged over all grid cells. The five experiments and the snow-depth satellite observations are compared to the IceBridge airborne snow-depth observations.»

8. Figure 8: Are these monthly averages over the 3 years ? The caption is hard to read because the "seven day" and "forecast" are too far apart.
Yes, they are, new caption: «RMSE of monthly averaged (over three years) ensemble mean of seven-day forecast SIC validated against a) AMSR-E/2 SIC observations and b) OSISAF SIC observations.»

9. Figure 9: Please specify in the caption that (b) is the monthly averaged RMSE over the three years.
This is now corrected, new text: «The figures show RMSE of the ensemble mean SIC averaged over 3 years and verified against the assimilated OSISAF SIC.»

10. Figure 10: I think it is unrealistic to use re-analysed forcing for the seasonal forecasts, as the re-analysed forcing would not be available in an operational real time context.
Agreed, this is of course not very realistic, but more a simplification since the focus of this study is on the assimilation. Since we compare our assimilation results with a control run using the same reanalysed forcing we think that the comparison is fair.

Technical corrections :

Page 5, line 8: Change ". . .observations where given..." to ". . .observations were given..."
This is corrected

Caption of Figure 3 : Change "low concentration ice (> 50 %)" to "low concentration ice (< 50 %)"
This is corrected

Caption of Figure 4 : Change "the blue stars the OSISAF" to "the red stars the OSISAF".
This is corrected

Caption of Figure 4 : For ice volume the units are km. Is it the volume per unit area ?
No, this is an error. The figures has been updated.

Caption of Figure 4 : It would be easier for the readers to mention that the x-labels are month-year.
Text added: The xlabel is given as month-year.

Page 14, line : Change "sea-ice extent being too large and the ice is too thick" to "sea-ice extent being too large and the ice being too thick".
This is corrected

Page 18, line 18: "The figures show that that . . ."
This is corrected

Figure 10: Change "The blue line represents a forecast using a climatological forcing made from an over of atmospheric data for 2000-2014 with assimilation" to "The blue line represents a forecast using a climatological forcing made from atmospheric data over 2000-2014 with assimilation"
This is corrected

Page 21, line 31: Change "The main parameters analysed in this study snow depth, SIT and SIC all vary on longer time scales than one week for the spatial resolution in our model" to "The main parameters analysed in this study, snow depth, SIT and SIC, all vary on longer time scales than one week for the spatial resolution in our model"
This is corrected

---

## Referee Comment (RC2) · Anonymous Referee #2 · 4 Dec 2018

General Comments

In this paper, the authors perform 5 experiments with a 20 km coupled ROMS-CICE model forced with ERA-Interim forcing for 3 full years for the period of 2011-2013. The five experiments are 1) assimilation of OSISAF sea ice concentration (SIC) only, 2) assimilation of OSISAF SIC and CryoSat-2 sea ice thickness (SIT), 3) assimilation of OSISAF SIC and SMOS SIT, 4) assimilation of OSISAF SIC and AMSR-E/2 snow depth observations, and 5) control run without any data assimilation. The Ensemble Kalman Filter (EnKF) is the data assimilation technique used in this study. Ocean boundary conditions are provided by the FOAM ocean model. Two sets of experiments

are performed: 1) assimilation experiments with 20 ensemble members with a 7-day assimilation time step, 2) seasonal forecasts with 20 ensemble members for the five-month period beginning in April/May to examine the skill in predicting the September sea ice minimum extent.

The authors computed the annual RMSE of the ensemble mean SIC over the three-year period and found that from January – August, the SIT experiments performed similarly and outperformed the SIC-only run during that period when using the weighted AMSR-E/2 data. From September – November, the SIC experiment had the lowest error. This could be related to no IT data during the summer months. The authors speculate the model has difficulty in transitioning from the melt to growing season. When comparing against the OSISAF ice concentration (which was assimilated into the model), the SIT experiment using SMOS showed the lowest RMSE from January – July. The snow depth experiment showed a lower RMSE than the SIC-only experiment for the period of January – June.

The authors examined "hit rates" to determine which experiment led to the most accurate number of grid cells classified as open water (concentration < 10%), low (<50%) or high concentration (>50%) and found that the experiments with the assimilation of ice thickness performed best. Total ice volume is examined for all 5 experiments and they find that except for the control run, the volume steadily decreases from year to year. The authors need to better address why this is happening, and propose future studies to investigate this further.

Comparisons are performed with the annual mean ice thickness and snow depth from all 5 experiments versus data from NASA Operation IceBridge. Since IceBridge data is only available for typically 10 transects for March/April each year; this is not a very compelling analysis. While Arctic snow depth data is difficult to obtain, it is recommended that the authors examine additional sources of ice thickness data, such as ice mass balance data (see comment below) which has much better temporal and spatial resolution.

Seasonal forecasts are evaluated by performing 5-month experiments for all five cases beginning in April of 2011, 2012 and 2013 to examine the SIC RMSE. When averaged for all three years, the SIT experiments perform best. Through mid-June, the snow depth experiment is very similar to the CryoSat-2 (SITI), but afterward the error increased significant and mirrors the control runs high error from August through September.

With the exception to the Lisæter (2007) reference, throughout the paper you should consistently refer to CryoSat as CryoSat-2.

Why aren't ice mass balance buoys used in your study? Look at available data at: http://imb-crrel-dartmouth.org/results/. During the period of your study, there is drifting buoy data available.

Are melt ponds used in your CICE simulations? If yes, state that in Section 4.3.

Why didn't you evaluate model ice drift errors using the International Arctic Buoy Programme buoy data? See http://iabp.apl.washington.edu/

This is a very well written paper with clear tables and complementary graphics. I recommend publication after my comments are addressed.

Specific Comments

Page 2 lines 15-25: Suggest you add the following reference to this section when discussing operational system assimilating SIC:

Posey, P. G., Metzger, E.J., Wallcraft, A.J., Hebert, D.A., Allard, R.A., Smedstad, O.M., Phelps, M.W, Fetterer, F., Stewart, J.S., Meier, W.N., Helfrich, S.R., 2015. Assimilating high horizontal resolution sea ice concentration data into the US Navy's ice forecast systems: Arctic Cap Nowcast/Forecast System (ACNFS) and the Global Ocean Forecast System (GOFS 3.1). The Cryosphere 9 2339-2365. doi: 10.5194/tcd-9-2339-2015.

Page 3 first paragraph: Consider adding the following recent references when discussing the use of CryoSat-2 data:

Allard, R. A., Farrell, S. L., Hebert, D. H., Johnston, W. F., Li, L., Kurtz, N. T., Phelps, M. W., Posey, P. G., Tilling, R., Ridout, A., and Wallcraft, A. L.: Utilizing CryoSat-2 sea ice thickness to initialize a coupled ice-ocean modeling system, Advances in Space Research, 62, doi:10.1016/j.asr.2017.12.030, 2018.

Blockley, E. W. and K. A. Peterson: Improving Met Office seasonal predictions of Arctic sea ice using assimilation of CryoSat-2 thickness, Cryosphere, 12, 3419–3438, doi:10.5194/tc-12-3419-2018.

Xie, J., F. Countillon, and L. Bertino: Impact of assimilating a merged sea-ice thickness from CryoSat-2 and SMOS in the Arctic reanalysis, Cryosphere, 12, 3671-3691, doi:10.5194/tc-12-3671-2018.

Page 4 line 12: Please state the horizontal resolution of the ERA-Interim dataset

Page 4 lines 13-14: You use FOAM for prescribed ocean boundary conditions. What do you use for the CICE boundary conditions?

Page 6 last paragraph: What is the accuracy of the AMSR-E/2 snow depth data?

Page 8: You state the coupled modeling system is run for 1 year as an initial state. Was it spun-up from rest? How was ice initialized? Uniform everywhere from a particular thickness?

Page 8 last paragraph: Why didn't you include another experiment which included a blended CryoSat-2/SMOS ice thickness?

Page 13 Figure 4b: Please explain your views on why the ice volume (except for control run) steadily decreases. I suggest in your conclusions section to include to some possible follow-on studies to better investigate this issue.

Page 14 lines 20-24: Please include figures and discussion on comparison for April

2012 and 2013?

Page 14 last paragraph: Have you looked at Dartmouth (formerly CRREL) IMB data for an additional source of ice thickness data? These data sets have much more temporal coverage than just Mar/Apr from NASA IceBridge.

Page 16 last paragraph: Table 3 shows yearly averaged RMSE values of ensemble average of snow depth compared to NASA IceBridge. Explain how you can do this when NASA IceBridge is only available for Mar/Apr each year.

Page 18 lines 15-16: You state five-month forecasts, but experiments are performed April – September What are the actual dates? Apr 30 – Sept 30 would be 5 months; April 1 – Sept 30 would be 6 months.

Technical Corrections:

Page 1 line 7: replace "asses" to "assess"

Page 1 line 12: should be CryoSat-2 (and throughout the paper)

Page 1 line 16: replace "lead" to "led"

Page 2 line 14: add comma after "later"

Page 8 line 30: reword "Five assimilation experiments" to "Five experiments"

Page 12: Fig 3 caption: first line should read "low concentration ice <50%" (not >50)

Page 13 line 8: replace "to much ice" to "too much ice"

Page 26 line 24: Provide more complete info for Sakov EnKF-C user guide (2015) reference

---

## Author Comment (AC2) · 17 Dec 2018

**Response comment Reviewer 2**

Anonymous Referee #2

General Comments

In this paper, the authors perform 5 experiments with a 20 km coupled ROMS-CICE model forced with ERA-Interim forcing for 3 full years for the period of 2011-2013. The five experiments are 1) assimilation of OSISAF sea ice concentration (SIC) only, 2) assimilation of OSISAF SIC and CryoSat-2 sea ice thickness (SIT), 3) assimilation of OSISAF SIC and SMOS SIT, 4) assimilation of OSISAF SIC and AMSR-E/2 snow depth observations, and 5) control run without any data assimilation. The Ensemble Kalman Filter (EnKF) is the data assimilation technique used in this study. Ocean boundary conditions are provided by the FOAM ocean model. Two sets of experiments are performed: 1) assimilation experiments with 20 ensemble members with a 7-day assimilation time step, 2) seasonal forecasts with 20 ensemble members for the fivemonth period beginning in April/May to examine the skill in predicting the September sea ice minimum extent.

The authors computed the annual RMSE of the ensemble mean SIC over the threeyear period and found that from January – August, the SIT experiments performed similarly and outperformed the SIC-only run during that period when using the weighted AMSR-E/2 data. From September – November, the SIC experiment had the lowest error. This could be related to no IT data during the summer months. The authors speculate the model has difficulty in transitioning from the melt to growing season. When comparing against the OSISAF ice concentration (which was assimilated into the model), the SIT experiment using SMOS showed the lowest RMSE from January – July. The snow depth experiment showed a lower RMSE than the SIC-only experiment for the period of January – June.

The authors examined "hit rates" to determine which experiment led to the most accurate number of grid cells classified as open water (concentration < 10%), low (<50%) or high concentration (>50%) and found that the experiments with the assimilation of ice thickness performed best. Total ice volume is examined for all 5 experiments and they find that except for the control run, the volume steadily decreases from year to year. The authors need to better address why this is happening, and propose future studies to investigate this further.

The decrease in sea-ice volume is not a model problem, but a response to the assimilation where the model, in general, has too much and too thick ice. An attempt at a discussion of this case was given on page 11. line 26-32 in the old manuscript. This section has now been modified to make this more clear, new text is added: « The control model is found to have too thick ice compared to the observations, while the experiments assimilating SIT are much closer to the observations, though largely biased. This can be used to explain the drastic decrease in sea-ice volume found in Fig.

\ref{fig:extent_volume}b. The model SIT is adjusting towards the observations by rapidly thinning the sea-ice.»

Comparisons are performed with the annual mean ice thickness and snow depth from all 5 experiments versus data from NASA Operation IceBridge. Since IceBridge data is only available for typically 10 transects for March/April each year; this is not a very compelling analysis. While Arctic snow depth data is difficult to obtain, it is recommended that the authors examine additional sources of ice thickness data, such as ice mass balance data (see comment below) which has much better temporal and spatial resolution.

Thank you for this valuable suggestion. We were not aware of these observations, and they are now included in the validation of the experiments.

Seasonal forecasts are evaluated by performing 5-month experiments for all five cases beginning in April of 2011, 2012 and 2013 to examine the SIC RMSE. When averaged for all three years, the SIT experiments perform best. Through mid-June, the snow depth experiment is very similar to the CryoSat-2 (SITI), but afterward the error increased significant and mirrors the control runs high error from August through September.

With the exception to the Lisæter (2007) reference, throughout the paper you should consistently refer to CryoSat as CryoSat-2.

This is corrected

Why aren't ice mass balance buoys used in your study? Look at available data at: http://imb-crrel-dartmouth.org/results/. During the period of your study, there is drifting buoy data available.

They are now included.

 Are melt ponds used in your CICE simulations? If yes, state that in Section 4.3.

Yes, the model use melt pond parametrization. Information regarding this is now added to the description of the model in section 2: «The model has a thermodynamic component calculating the local growth rate of snow and ice, ice dynamics component calculating ice drift based on the material ice strength, a transport component, a melt pond parametrization and a ridging parametrization used to distribute ice in thickness categories \citep{Hunke_2015}.»

Why didn't you evaluate model ice drift errors using the International Arctic Buoy Programme buoy data? See http://iabp.apl.washington.edu/

We did not have in mind to include ice drift in the study, thus unfortunately, the model drift output was not saved.

This is a very well written paper with clear tables and complementary graphics. I recommend publication after my comments are addressed.

We thank the reviewer for the kind words and the careful and constructive feedback.

Specific Comments

Page 2 lines 15-25: Suggest you add the following reference to this section when discussing operational system assimilating SIC:

Posey, P. G., Metzger, E.J., Wallcraft, A.J., Hebert, D.A., Allard, R.A., Smedstad, O.M., Phelps, M.W, Fetterer, F., Stewart, J.S., Meier, W.N., Helfrich, S.R., 2015. Assimilating high horizontal resolution sea ice concentration data into the US Navy's ice forecast systems: Arctic Cap Nowcast/Forecast System (ACNFS) and the Global Ocean Forecast System (GOFS 3.1). The Cryosphere 9 2339-2365. doi: 10.5194/tcd-9-2339- 2015.

Thank you for the advice, this has now been added: «\cite{posey2015assimilating} assimilated high-resolution SIC observations (~4 km) into a coupled ocean sea-ice model, the Arctic Cap Nowcast/Forecast System (ACNFS) using the 3DVAR assimilation method. In this study, they showed that increased observation resolution has a significant impact on the ice-edge forecast.»

Page 3 first paragraph: Consider adding the following recent references when discussing the use of CryoSat-2 data:

Allard, R. A., Farrell, S. L., Hebert, D. H., Johnston, W. F., Li, L., Kurtz, N. T., Phelps, M. W., Posey, P. G., Tilling, R., Ridout, A., and Wallcraft, A. L.: Utilizing CryoSat-2 sea ice thickness to initialize a coupled ice-ocean modeling system, Advances in Space Research, 62, doi:10.1016/j.asr.2017.12.030, 2018.

Blockley, E. W. and K. A. Peterson: Improving Met Office seasonal predictions of Arctic sea ice using assimilation of CryoSat-2 thickness, Cryosphere, 12, 3419–3438, doi:10.5194/tc-12-3419-2018.

Xie, J., F. Countillon, and L. Bertino: Impact of assimilating a merged sea-ice thickness from CryoSat-2 and SMOS in the Arctic reanalysis, Cryosphere, 12, 3671-3691, doi:10.5194/tc-12-3671-2018.

Thank you, we were not aware of the recent papers. We have improved the text to include these studies: «In the last couple of years, there has also been an increase in the use of Cryosat-2 observations in various forms for assimilation. \cite{Chen_2017} assimilated both the SMOS thin SIT and the CryoSat-2 thick SIT into the National Centers for Environmental Prediction's (NCEP) Climate Forecast System version 2 \citep{Saha_2014} using the localized error subspace transform

ensemble Kalman filter \citep{Nerger_2013}. This study showed improved sea-ice prediction with SIT assimilation, thus verifying the importance of SIT observations to achieve accurate sea-ice forecasts. \cite{xie2018impact} assimilated a blended SMOS CryoSat-2 product into TOPAZ. They showed that these observations provide the primary source of observational information in the central Arctic, and when assimilating this product the model SIT was improved. \cite{blockley2018improving} argued that by assimilating Cryosat-2 observations, the Arctic summer prediction of ice extent and location were significantly improved. \cite{allard2018utilizing} used CryoSat-2 observations for initialization in the coupled ocean sea-ice ACNFS model. The study showed improved model thickness with CryoSat-2 initialization when compared to independent ice thickness observations.»

Page 4 line 12: Please state the horizontal resolution of the ERA-Interim dataset

This is now added: «The coupled model is forced by atmospheric data from the ERA-Interim dataset from the European Centre for Medium Ranged Weather Forecast \citep[ECMWF; ][] {Dee_2011}.  The ERA-Interim dataset has a horizontal resolution of approximately 0.7$^\circ$, corresponding to T255 spectral truncation.»

Page 4 lines 13-14: You use FOAM for prescribed ocean boundary conditions. What do you use for the CICE boundary conditions?

At the moment no boundary conditions are used for the sea-ice. For the most part of the year, this is not a problem because the sea-ice is surrounded by ocean. While during winter when the sea-ice extends beyond the Behring straight this might be a problem for this area, but this is not included here, and we do not believe this to be an issue regarding the results.

Page 6 last paragraph: What is the accuracy of the AMSR-E/2 snow depth data?

This has now been added: «For the snow depth product uncertainty estimates exist for every grid point. There are two main sources of uncertainties in this observation product: The first is that the number of IceBridge observations used to develop the empirical relationship between brightness temperatures and snow depths is small compared to the coverage of the product. The second uncertainty is in the input parameters (brightness temperature, ice concentration etc.). More on how the uncertainties are explicitly calculated can be found in \citep{Rostosky_2018}.»

Page 8: You state the coupled modeling system is run for 1 year as an initial state. Was it spun-up from rest? How was ice initialized? Uniform everywhere from a particular thickness?

This has now been clarified in the manuscript: «The initial ensemble is generated from ice states from January 1. based on 20 different years of a standalone ice model run without assimilation. The standalone model was initialised without ice in 1979. All initial ocean states are model output at initial date January 1. 2010. This output is taken from a model spin-up over1993-2010.»

Page 8 last paragraph: Why didn't you include another experiment which included a blended CryoSat-2/SMOS ice thickness?

The focus of this study was on the individual observation products. , we could have tested all sorts of combination between the products, but this would become quite messy. In addition, the blended product provided an alternative dataset for model verification.

Page 13 Figure 4b: Please explain your views on why the ice volume (except for control run) steadily decreases. I suggest in your conclusions section to include to some possible follow-on studies to better investigate this issue.

This is already mentioned on p.11 lines 26-32 in the previous version of the manuscript. The decrease is related to too much ice in the control model, and due to assimilation, the thickness is slowly going towards the observed values which are thinner. This section has been updated to further clarify this result, see comment above.. Thus no further studies of this effect should be necessary.

Page 14 lines 20-24: Please include figures and discussion on comparison for April 2012 and 2013?

Figure 7. is only meant as an illustration of the curves shown in figure 4b. Even though the figures could be added for 2012 and 2013 we do not think this would provide any additional information from figure 4b. Also, 8 more figures would take a lot of space in the manuscript and would be too messy as they are providing only little new information.

Page 14 last paragraph: Have you looked at Dartmouth (formerly CRREL) IMB data for an additional source of ice thickness data? These data sets have much more temporal coverage than just Mar/Apr from NASA IceBridge.

Yes, as mentioned previously a new section including verification of these data have been added: « Another independent dataset of SIT observations complementing the IceBridge observations by observations throughout the year is the IMB buoy dataset. The result of model validation with the IMB is shown in table \ref{tab:IMB_th}. For these observations, a slightly different method than that applied for IceBridge is performed. This is because IceBridge temporarily only covered March-April, while the IMB data span the entire year. The buoy observations are converted to daily averages on the model grid. From these values, the RMSE is calculated on the 7-day ensemble mean and averaged for each year. Since SIT is a relatively slow varying variable, for each 7-day output, observations from +/-3 days are used to increase the number of observations. The IMB observations do not include an uncertainty estimation, hence the RMSE is not normalised was the case for other other RMSE values in this work. The results show that over the three study years, the SMOS internal SIT assimilation system has the lowest RMSE values, followed by the CryoSat-2 internal SIT assimilation. The other three show similar results, indicating the positive impact of assimilating ice thickness in the model.» A table has also been added illustrating the yearly averaged results.

Page 16 last paragraph: Table 3 shows yearly averaged RMSE values of ensemble average of snow depth compared to NASA IceBridge. Explain how you can do this when NASA IceBridge is only available for Mar/Apr each year.

The yearly average is a Mar/Apr average. This has now been made clearer by adding it into the figure text: «The Mar/Apr-mean RMSE of the ensemble-mean snow depth averaged over all grid cells.» In addition, new text has been added: The same method as for the SIT in table \ref{tab:IceBridge} was used, where Mar/Apr model values are compared to the IceBridge observations and averaged.

Page 18 lines 15-16: You state five-month forecasts, but experiments are performed April – September What are the actual dates? Apr 30 – Sept 30 would be 5 months; April 1 – Sept 30 would be 6 months.

We mean it is an approximately 5-month forecast. The start date varied a bit because of the 7-day assimilation cycle. Information regarding this is now added to the text: «This is done by running each of the experiments from mid-April to mid-September each year without assimilation and validating against the OSISAF SIC observations. The actual start date varied slightly from year to year because of the 7-day assimilation cycle, but the start date was the same for all experiments.»

Technical Corrections:

Page 1 line 7: replace "asses" to "assess"

done

Page 1 line 12: should be CryoSat-2 (and throughout the paper)

done

Page 1 line 16: replace "lead" to "led"

done

Page 2 line 14: add comma after "later"

done

Page 8 line 30: reword "Five assimilation experiments" to "Five experiments"

done

Page 12: Fig 3 caption: first line should read "low concentration ice <50%" (not >50)

done

Page 13 line 8: replace "to much ice" to "too much ice"

done

Page 26 line 24: Provide more complete info for Sakov EnKF-C user guide (2015) reference

Added arXiv: «Sakov, P.: EnKF-C user guide. arXiv:1410.1233., 2015.